# Single-trial cross-area neural population dynamics during long-term skill learning

T. L. Veuthey[1,2,3,4,5], K. Derosier [1,3,4,5], S. Kondapavulur [2,3,4] & K. Ganguly [3,4 ✉]

Mammalian cortex has both local and cross-area connections, suggesting vital roles for both local and cross-area neural population dynamics in cortically-dependent tasks, like movement learning. Prior studies of movement learning have focused on how single-area population dynamics change during short-term adaptation. It is unclear how cross-area dynamics contribute to movement learning, particularly long-term learning and skill acquisition. Using simultaneous recordings of rodent motor (M1) and premotor (M2) cortex and computational methods, we show how cross-area activity patterns evolve during reach-to-grasp learning in rats. The emergence of reach-related modulation in cross-area activity correlates with skill acquisition, and single-trial modulation in cross-area activity predicts reaction time and reach duration. Local M2 neural activity precedes local M1 activity, supporting top–down hierarchy between the regions. M2 inactivation preferentially affects cross-area dynamics and behavior, with minimal disruption of local M1 dynamics. Together, these results indicate that cross-area population dynamics are necessary for learned motor skills.

[1] Neuroscience Graduate Program, University of California San Francisco, San Francisco, CA, USA. [2] Medical Scientist Training Program, University of California San Francisco, San Francisco, CA, USA. [3] Neurology and Rehabilitation Service, San Francisco Veterans Affairs Medical Center, San Francisco, CA, USA. [4] Department of Neurology, University of California San Francisco, San Francisco, CA, USA. [5] These authors contributed equally: T. L. Veuthey, K. Derosier. ✉email: karunesh.ganguly@ucsf.edu

The connectivity pattern of mammalian cortex, characterized by both local and cross-area connections[1], suggests an important role for interactions between population dynamics compartmentalized locally and those coordinated between regions. But it is unknown whether population dynamics coordinated across multiple cortical areas contribute to long-term skill learning. In the motor system, it has been shown that both premotor cortex (M2)[2–6] and motor cortex (M1)[7–11] demonstrate changes in local population dynamics with motor learning. However, it remains unclear how cross-area dynamics between M1 and M2 are coordinated and change with long-term skill learning. Previous work on cross-area interactions during motor learning has focused on macroscopic population activity, such as local field potentials[12–14] and wide-field calcium signals[4,15]. However, such measures of aggregate activity collapse signals from a heterogeneous population of neurons into a single signal, making it difficult to resolve potentially important multiplexed signals within that population[16–18]. Recent work has examined cross-area dynamics during motor adaptation[5], but this process is fundamentally different from new skill learning[19].

How can we examine cross-area population dynamics during learning, especially when newly learned movements are still variable? To avoid the limitations of analyzing trial-averaged movement-related signals, we can instead build models by estimating prevalent population patterns from signals concatenated over time[17,20,21]. One common approach in well-trained animals is to use dimensionality reduction methods such as Principal Component Analysis (PCA) to capture patterns of dominant covariance within local populations[5,7,11,16,22–29]. Those reduced local signals can then be compared across regions[5,16,30]. However, since PCA finds dimensions that maximize local variance, activity patterns which do not dominate local variance are discarded. Thus, this approach may dismiss as noise neural fluctuations representing activity coordinated across areas[31]. Instead, cross-area activity might be identified by directly detecting covariance which is coordinated across regions[5,18,32]. Recent work has shown that simultaneous recordings from two visual areas can be analyzed to identify a neural shared subspace defined by the activity in each region that is maximally correlated with activity in a partner region[32]. Two additional studies have also identified widespread neural signals encoding facial movements[18] and thirst-based motivational states[33]. These findings suggest that signals shared across brain areas may contribute to coordinating diverse behaviors. But it remains unclear whether and how cross-area dynamics evolve during learning. Understanding these changes can help define the functional role of cross-area activity, and provide new insight into learning mechanisms of distributed networks.

Here, we aim to assess how population dynamics shared by M2 and M1 change during motor skill learning. We hypothesize that M2–M1 shared dynamics coordinate information between the regions and contribute to learning complex behaviors. To isolate activity shared across areas, we perform simultaneous multisite recordings in M2 and M1 and use the dimensionality reduction technique Canonical Correlation Analysis (CCA) to define the axes of maximal correlation between the M2 and M1 neural populations[34]. By simultaneously reducing dimensionality and optimizing for M2–M1 correlation, CCA can identify cross-area signals that may be missed by methods that exclusively optimize local variance. We use the term cross-area to refer to activity in each area which is maximally correlated with activity in the partner region. We thus aim to explicitly identify cross-area dynamics during both early exploratory learning and late learned execution of a skilled movement.

In each region, we find that cross-area dynamics modulation is proportional to single-trial reaching behavior, and that modulation to reach initiation and reach duration is amplified with learning. We additionally find that local activity in M2 precedes local activity in M1, consistent with a top–down hierarchy between the signals more specific to M2 and M1. In line with this top–down functional role, M2 inhibition in well-trained animals impairs reach behavior and disrupts reach representation in M1 cross-area signals. Together, our results indicate that cross-area M2–M1 population dynamics represent a necessary component of skilled motor learning.

## Results

**Learning increases movement-modulated neurons in M1 and M2.** We performed simultaneous recordings of population neural activity in M2 and M1 (Fig. 1a, Supplementary Fig. 1) in rats learning a cue-driven reach-to-grasp task, a well-established model for skill learning[27,35,36]. Both M2 and M1 are required for learning and execution of reach-to-grasp movements in both rodents and primates[37–39]. Animals learned to successfully retrieve pellets with training (hierarchical bootstrap, $10^4$ shuffles used here and hereafter, 27.28% ± 3.06 for Early, 57.64% ± 2.49 for Late, $p < 0.0001$, $n = 5$ rats). There were concomitant improvements in movement duration (hierarchical bootstrap, 0.30 s ± 0.056 for Early, 0.20 s ± 0.040 for Late, $p = 0.0027$, $n = 5$ rats) and reaction time (hierarchical bootstrap, 32.23 s ± 24.58 for Early, 0.89 s ± 0.18 for Late, $p < 0.0001$, $n = 5$ rats) (Fig. 1b, Supplementary Fig. 2, Supplementary Table 1).

To examine relationships between single-neuron activity and movements, we created trial-averaged peri-event time histograms (PETHs) for both M2 and M1 in early and late learning. We used a circular shuffle test to quantify whether each neuron was significantly modulated ($p < 0.000125$) (Fig. 1c, d; Methods)[40]. Over learning, significantly more neurons in both areas were movement modulated (hierarchical bootstrap, $n = 5$ rats; M1: 59.83% ± 8.89 for Early, 94.32% ± 4.65 for Late, $p < 0.0001$; M2: 48.19% ± 13.40 for Early, 88.03% ± 5.81 for Late, $p < 0.0001$), consistent with prior work arguing that learning engages and amplifies representations in both regions[4,10,27,41,42]. However, as PETHs represent neural activity averaged across trials, increased PETH modulation can be driven by many neural and behavioral factors. In addition, trial-averaged activity from both early and late learning demonstrated evidence of sequential activation of neurons in a task-dependent manner[10,17,21,40,43,44]. While visual inspection of M1 and M2 PETHs in early and late learning suggest changes in correlated firing between areas, trial-averaged data, such as PETHs, inherently blur trial-by-trial variations in neural activity which may correspond to trial-by-trial variations in performance. Therefore, although PETHs from M1 and M2 may show temporal overlap, this does not indicate single-trial correlated activity. PETHs also require time locking to a specific aspect of movement. Thus, further analyses of single-trial data are essential for distinguishing between these confounding variables.

Without modeling single-trial activity patterns, it is unclear how movement signals in M2 and M1 correspond with single-trial performance over learning, or how task-relevant activity is coordinated between M1 and M2 on a moment-by-moment basis. Moreover, studies on population dynamics have identified that single-trial dynamics can be reliable and revealing about single-trial behavioral variation in well-trained animals[18,28,45,46]. However, these findings raise the fundamental questions of whether single-trial cross-area dynamics can inform our understanding of cortical communication during learning.

**Distinct cross-area dynamics versus single-unit interactions.** How then can we identify single-trial activity patterns shared between M2 and M1? We used CCA, which finds linear

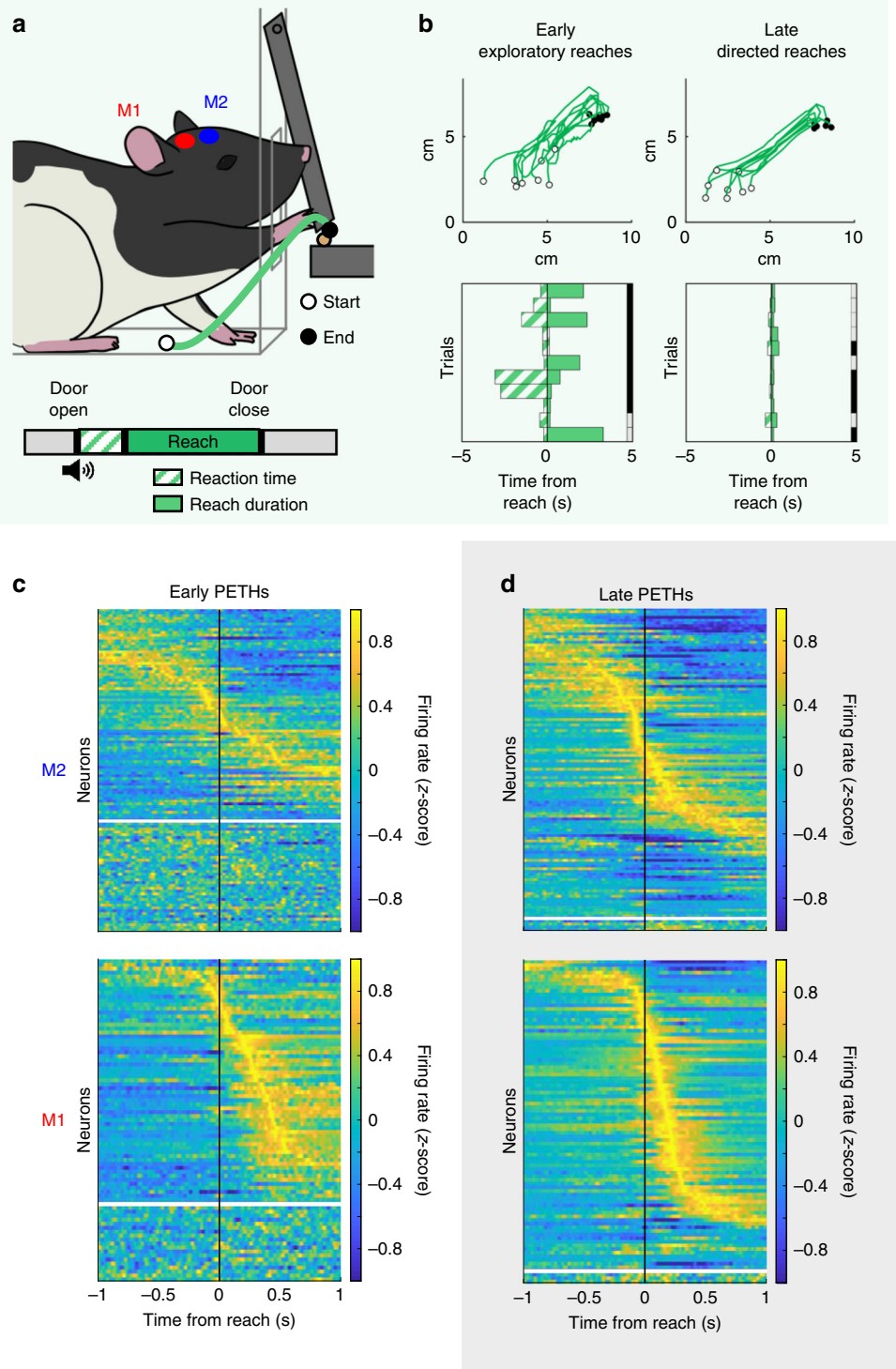

**Fig. 1 Motor skill learning is associated with increased cortical movement signals. a** (Top) Rats were trained to perform the reach-to-grasp task. (Bottom) Single-trial experimental paradigm. **b** Example reaches in early (left) and late (right) learning. (Top) Paw trajectories. White dot marks reach start position. Black dot marks reach end position. (Bottom) Example consecutive single-trial representations of reaction time (green striped bars) and reach duration (green bars). Right border of plot shows accuracy, with pellet retrieval success in gray and failure in black. **c** Population trial-averaged peri-event time histograms (PETHs) for premotor cortex (M2) (top) and primary motor cortex (M1) (bottom) units in early learning ($n = 5$ rats). Significantly modulated neurons are shown above the white line, ordered by the time of their peak modulation. Nonsignificantly modulated neurons are shown below the white line, ordered by channel number. Firing rates are z-scored per-neuron for display only. **d** As in **c**, but for late learning.

combinations of simultaneous M2 and M1 activity that are maximally correlated with each other, to measure cross-area dynamics. The neuron weights obtained using CCA define axes in the high-dimensional M1 and M2 population spaces along which activity is most similar (see Methods). The projections of high-dimensional neural activity onto these axes provide a low-dimensional representation of shared signals (Fig. 2a, b).

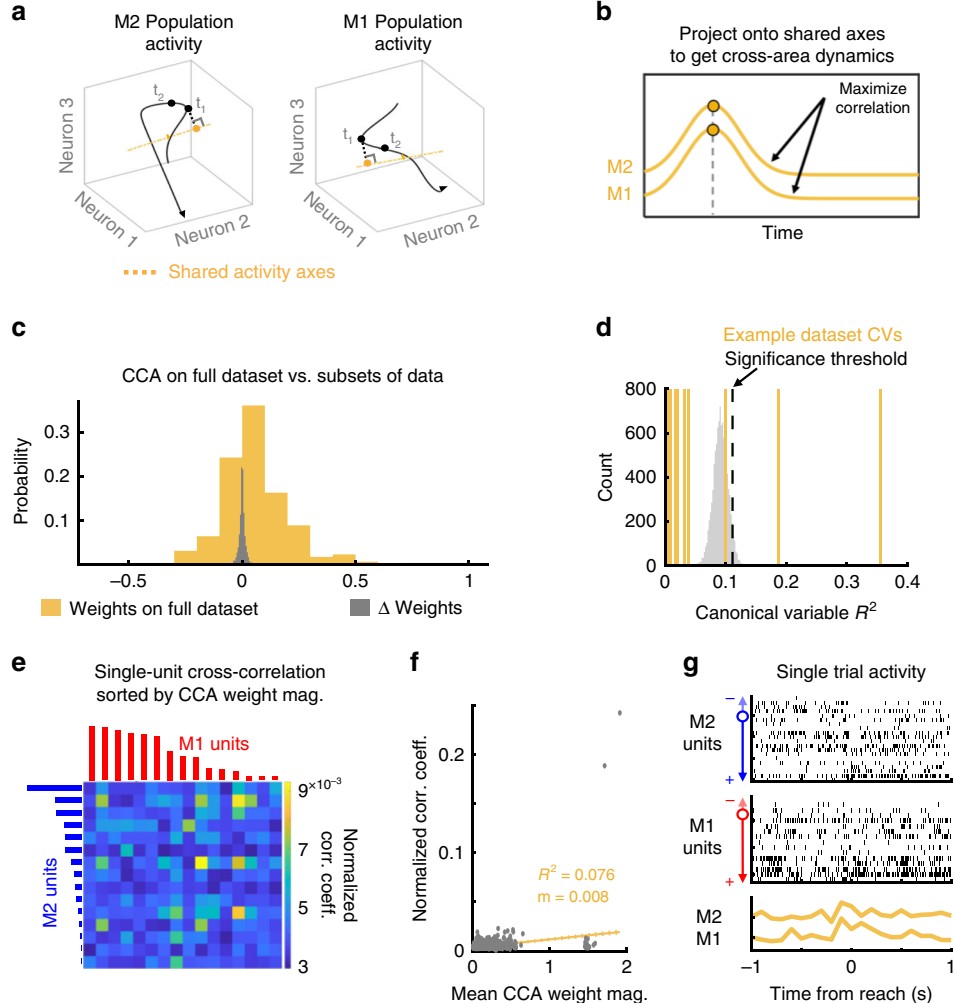

**Fig. 2 Canonical Correlation Analysis identifies shared cross-area dynamics. a** Illustration of method for identifying cross-area dynamics. A multidimensional neural space can be defined using the activity of each M2 (left) or M1 (right) neuron as one dimension. Neural trajectories are shown in black (artificial data). Canonical correlation analysis (CCA, yellow) is used to identify shared axes, such that when the neural trajectories are projected onto these axes, as shown in **b**, the resulting trajectories, called cross-area dynamics, are maximally correlated between M2 and M1. The yellow dots for M1 and M2 illustrations are the projected values for the same timepoint in **a**. **c** In yellow, distribution of CCA weights when fitting on the full datasets. In gray, distribution of weights differences from ten subsamplings of each dataset (i.e., weight from subset − weight from full dataset). **d** Example identification of significant canonical variables (CVs, yellow lines), relative to trial-shuffled data (gray distribution, $10^4$ shuffles). Significance threshold at 95th percentile of reference distribution. For this dataset, two CVs were significant. **e** Example comparison of CCA weight magnitude to single-neuron pairwise cross-correlations. M1 units (red) and M2 units (blue) are ordered by the absolute value of their CCA weight for the top CV. Color in the cross-correlogram indicates normalized peak correlation coefficient for timelags between −200 and +200 ms. **f** Across animals, pairwise cross-correlation is correlated with mean CCA weight magnitude for that neuron pair. **g** CCA identifies moment-to-moment shared covariation patterns that may be hard to see by eye in single-trial data. (Top) Population raster for an example trial in M2, where each row is a single neuron, sorted by CCA weight. (Middle) Population raster for the same trial in M1, sorted by CCA weight. (Bottom) The M2 and M1 cross-area dynamics for that same trial. The $R^2$ between the M2 and M1 cross-area dynamics on this example trial is 0.3733.

Our main analyses are done with CCA fit to neural data binned at 100 ms, with no timelag between regions. For comparison, we also fit models to data binned at 75 and 50 ms at timelags from −500 to +500 ms. We found that for most datasets, models fit using 100 ms bins with no timelag resulted in the best generalizability to held-out data (see Methods and Supplementary Fig. 3). In addition, we found that neuron weights and axes generated by CCA are different from those found with PCA, which instead defines axes of maximal variance in single-area population spaces (Supplementary Fig. 4). In each region, the angles between axes of maximal local covariance (using PCA) and axes of maximal cross-area correlation (using CCA) are significantly different from zero, and did not change with learning (hierarchical bootstrap, $n = 4$ rats; M2: 59.66° ± 4.57 for Early, 59.34° ± 3.83 for

Late, two-sided $p = 0.92$; M1: 49.84° ± 5.49 for Early, 59.47° ± 8.68 for Late, two-sided $p = 0.43$; in all cases, all bootstrap samples were >30°, $p < 0.0001$).

To validate the stability of this CCA axis, we calculated ten sets of CCA neuron weights from ten randomly selected subsets of 90% of timebins (held-out data was nonoverlapping). Across all datasets, the range of weights for models using the full datasets (mean ± std = 0.057 ± 0.16 a.u.) was much larger than the variance in each neuron's weights between subsets (weight from subset − weight from full dataset, mean ± std = −2.37e−05 ± 0.02 a. u.). That CCA weights change by a small amount when fit to different subsets of data suggests that the CCA model for M2–M1 cross-area activity is robust (Fig. 2c). Further analyses were conducted without data subsampling.

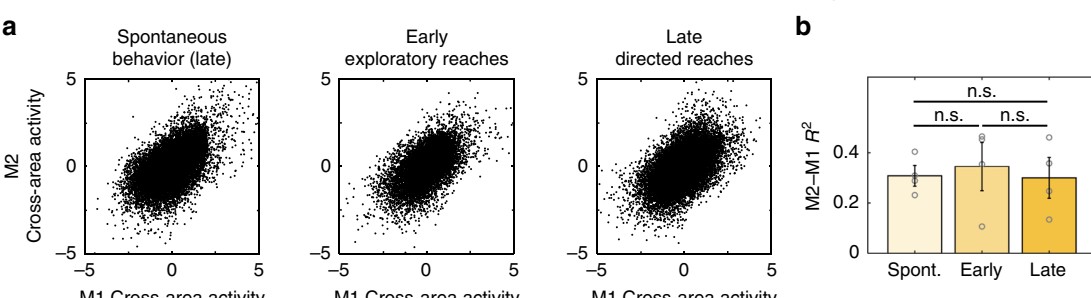

**Fig. 3 Correlation of M1–M2 cross-area activity is stable across behaviors. a** Correlation between M2 and M1 components of the M2–M1 cross-area population activity during (Left) spontaneous behavior, (Middle) early exploratory reaches, and (Right) late directed reaches. Spontaneous behavior was during the late learning day. Each data point is M2 and M1 data from a single 100 ms bin ($n = 4$ rats). **b** Quantification of (**a**) as crossvalidated $R^2$ values. Bars show mean ± SEM, open circles show data from individual animals ($n = 4$ rats). Correlation is not significantly different during spontaneous behavior, early reaches, and late reaches. Mixed-effect model, two-sided $t$-test, not adjusted for multiple comparisons.

To verify that the CCA-defined M2 and M1 cross-area activity models represented behaviorally significant coordinated activity, we compared the $R^2$ between the cross-area activity to CCA models fit on trial-shuffled data. We generated a reference distribution of $R^2$ values between the top M1 and M2 canonical variables (CVs) from $10^4$ iterations of trial-shuffled data. CVs fit to the true dataset were considered significant if their $R^2$ value exceeded the 95th percentile of the reference distribution (see Methods). Shuffling data between trials while maintaining within-trial temporal structure preserved and controlled for coarse activity fluctuations due to movement. We found that most datasets had one to three significant CVs (Fig. 2d), confirming that CCA identified low-dimensional activity shared across M2 and M1. For one animal, the early dataset had no significant CVs; this animal was excluded from further analysis.

Finally, we examined whether CCA identifies cross-area relationships equivalent to those identified using a single-neuron functional connectivity measure, short-latency cross-correlations[13,44,47,48]. Specifically, if CCA consistently assigns high weights to M2 and M1 neurons which also have high cross-correlation values, this would indicate that CCA finds M2–M1 activity shared by individual neurons in each region. However, if the CCA weights of M2 and M1 neurons do not vary with cross-correlation values, then CCA-defined activity instead reflects distributed shared population dynamics not obvious at the single-neuron level. We found there was a weak but significant correlation between the mean CCA weights of M2–M1 neuron pairs and their short-latency normalized cross-correlation values (linear regression, $R^2 = 0.0761$, $p = 5.98 \times 10^{-89}$, $t = 20.399$, $n = 5056$ neuron pairs from four rats) (Fig. 2e, f). Therefore, M2 and M1 CCA neuron weights capture aspects of short-latency correlations but can also capture additional information about cross-area dynamics (Fig. 2g). This suggests that communication between cortical areas maybe be better modeled based on population-wide activity patterns rather than based on interactions between single neurons.

**Correlation of cross-area activity is stable across learning.** Does learning change the correlation of cross-area dynamics? If learning simply increases M1–M2 activity coordination, we would expect the correlation of M1 and M2 cross-area activity to be lower during early exploratory actions than during skilled behavior. To address this, we correlated M1 and M2 cross-area activity during three types of behavior: spontaneous behavior, exploratory reaches in early learning, and directed reaches in late learning (Fig. 3a). Surprisingly, there was no difference in the mean correlation values ($R^2$) of M1 versus M2 cross-area activity during

the different behaviors (mixed-effect model, $0.31 \pm 0.04$ for Spontaneous, $0.34 \pm 0.10$ for Early, $0.30 \pm 0.08$ for Late; Spontaneous vs. Early: $t(6) = 0.46$, $p = 0.66$; Spontaneous vs. Late: $t(6) = -0.15$, $p = 0.89$; Early vs. Late: $t(6) = -0.74$, $p = 0.49$; $n = 4$ rats) (Fig. 3b). Thus, generally increased coordination between M2 and M1 activity by itself seems unlikely to drive performance gains.

**Learning amplifies cross-area encoding of reach initiation.** An intriguing alternative is that learning is due to the modification of task representations within cross-area dynamics. Specifically, signals within the existing range of cross-area activity may be remapped to represent information about the task. Thus, while the overall range of M1–M2 cross-area signals may not change, high amplitude cross-area activity may now be associated with a particular behavioral state. As noted above, we observed that the door open cue was more rapidly followed by reach initiation after learning (Fig. 1b), suggesting that the timing of reach initiation might be an important marker of learning. We thus explored whether M1–M2 cross-area activity could account for this change. To visualize this possibility, we plotted the M1 cross-area activity versus the M2 cross-area activity during pre-reach and at reach initiation (Fig. 4a). The histograms show the probability density functions of the respective subspace activity before and during reach initiation. Interestingly, the two behavioral states were significantly more separable after learning (mixed-effect model, M2: $0.31 \pm 0.15$ for Early, $1.29 \pm 0.18$ for Late, $t(6) = 5.3806$, $p = 0.0017$; M1: $0.27 \pm 0.14$ for Early, $1.09 \pm 0.12$ for Late, $t(6) = 6.7227$, $p = 0.00053$; $n = 4$ rats) (Fig. 4b), suggesting that high amplitude cross-area activity gained task relevance with learning.

The increased task relevance of high amplitude cross-area activity was also apparent on a single-trial basis. When viewed as single-trial trajectories, peaks in cross-area activity became associated with reach initiation after learning (Fig. 4c). We quantified this association across trials by building a logistic regression model to distinguish cross-area activity during two seconds before reaching versus a 400 ms window at reach initiation. Strikingly, detection of reach initiation based on this cross-area activity model improved with learning (mixed-effect model, $0.66 \pm 0.03$ for Early, $0.87 \pm 0.02$ for Late, $t(6) = 9.77$, $p = 6.63 \times 10^{-5}$, $n = 4$ rats) (Fig. 4d). Using the logistic regression model, we could then probe the time course of reach initiation prediction based on M1–M2 cross-area activity (Fig. 4e). On average, while the time of reach initiation was not well predicted during early trials, it became highly predictable after learning

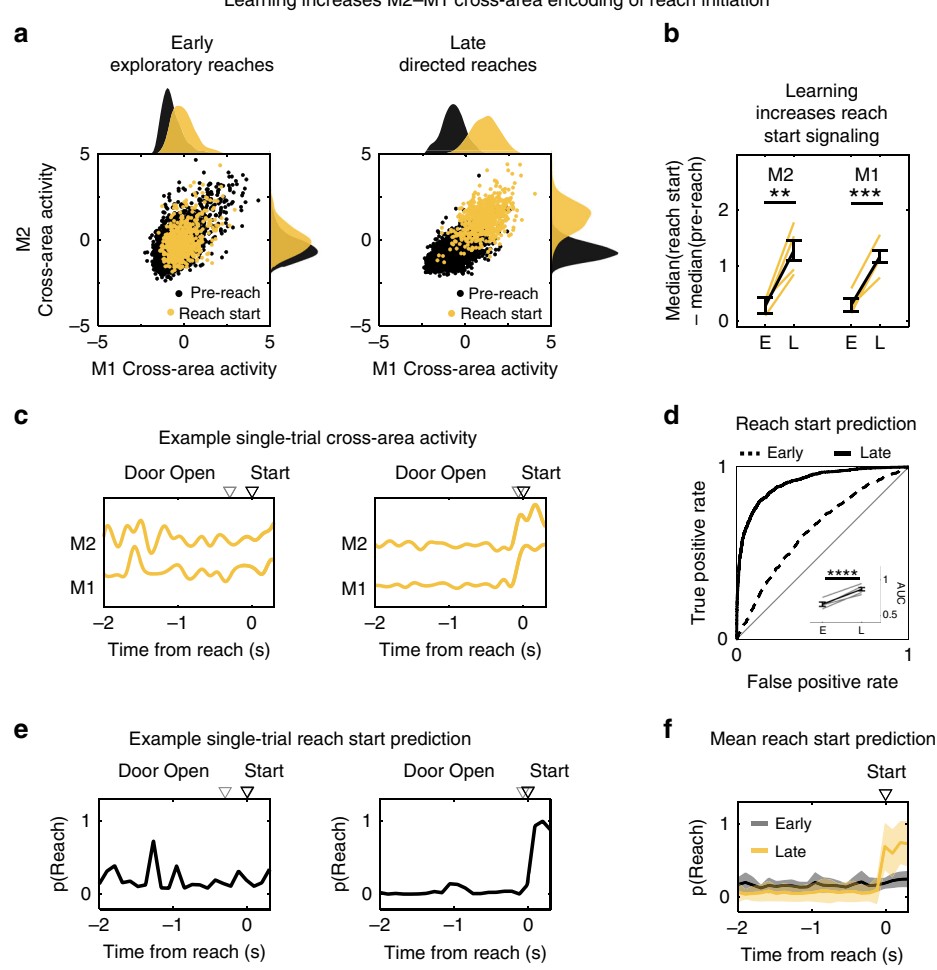

**Fig. 4 Learning increases cross-area representation of reach initiation. a** M2–M1 cross-area activity before reach and during reach initiation for example animal, in early (left) and late (right) learning. Probability density functions of M1 (top) and M2 (right). Pre-reach activity in black. Reach start activity in yellow. **b** Quantification of (**a**) as the difference between pre-reach and reach median activity during early and late learning for M2 ($p = 0.0017$) (left) and M1 ($p = 0.00053$) (right) cross-area activity. Yellow lines show values for individual animals, black line shows mean ± SEM, $n = 4$ rats. **$p < 0.01$; ***$p < 0.0001$; two-sided $t$-statistics, not adjusted for multiple comparisons. **c** Example single-trial M2 and M1 cross-area activity before and during reach initiation. (Left) Early learning. (Right) Late learning. Time to reach initiation is indicated by triangles marking door opening (gray open triangle) and reach start (black open triangle). **d** ROC analysis of detection of reach initiation from M2 and M1 cross-area activity using logistic regression (example animal). (Inset) Difference in reach detection with learning quantified as the area under the curve (AUC) for all animals ($p = 6.63 \times 10^{-5}$). Gray lines show values for individual animals, black lines show mean ± SEM, $n = 4$ rats. ****$p < 0.0001$. two-sided $t$-statistic, not adjusted for multiple comparisons. **e** Example single-trial prediction of reach initiation using the models built in **d**, same trials as **c**. (Left) Early learning. (Right) Late learning. **f** Comparison of mean prediction of reach initiation during early (gray) and late (yellow) learning for example animal. Shaded region shows SEM, $n = 4$ rats.

(hierarchical bootstrap, $0.12 \pm 0.044$ for Early, $0.30 \pm 0.082$ for Late, $p < 0.0001$, $n = 4$ rats) (Fig. 4f).

**Learning amplifies cross-area encoding of reach duration**. Does M1–M2 cross-area activity only coordinate movement initiation, or does it affect other aspects of reach performance? We examined whether single-trial M2–M1 cross-area dynamics were informative about single-trial reach duration, and whether the reach modulation of cross-area activity for movements of similar duration changed with learning. To quantify reach modulation in single-trial activity, we calculated a cross-area modulation metric (CA-modulation), which compared neural activity during reaching versus an equivalent baseline period for each trial (Fig. 5a, b). This measure is equivalent to the $d'$ (d-prime) signal sensitivity index used in signal processing (see Methods). To directly test the relationship between behavioral performance and M1 and M2 CA-modulation, we correlated CA-modulation with

reach duration on a trial-by-trial basis (Fig. 5c). Interestingly, we found that CA-modulation reliably covaried with reach duration, indicating that cross-area dynamics represent information relevant to behavioral performance (mixed-effect model, M2: log slope = $-0.27$, $t(1531) = -14.43$, $p = 2.36 \times 10^{-44}$; M1: log slope = $-0.23$, $t(1562) = -13.88$, $p = 2.05 \times 10^{-41}$; $n = 4$ rats). In addition, both M1 and M2 CA-modulation increased with learning (hierarchical bootstrap, $n = 4$ rats; M1: $0.45 \pm 0.13$ for Early, $1.97 \pm 0.44$ for Late, $p < 0.0001$; M2: $0.46 \pm 0.12$ for Early, $2.57 \pm 0.62$ for Late, $p < 0.0001$; Fig. 5d). This modulation increase was not simply due to overall improved reach performance; instead, movements of similar duration in early and late learning were more modulated in late learning, indicating amplified representation of learned skills. Thus, the process of learning appeared to enhance reach-specific signals in cross-area population dynamics. Amplification of these task-specific signals spanning multiple brain areas may be a mechanism for

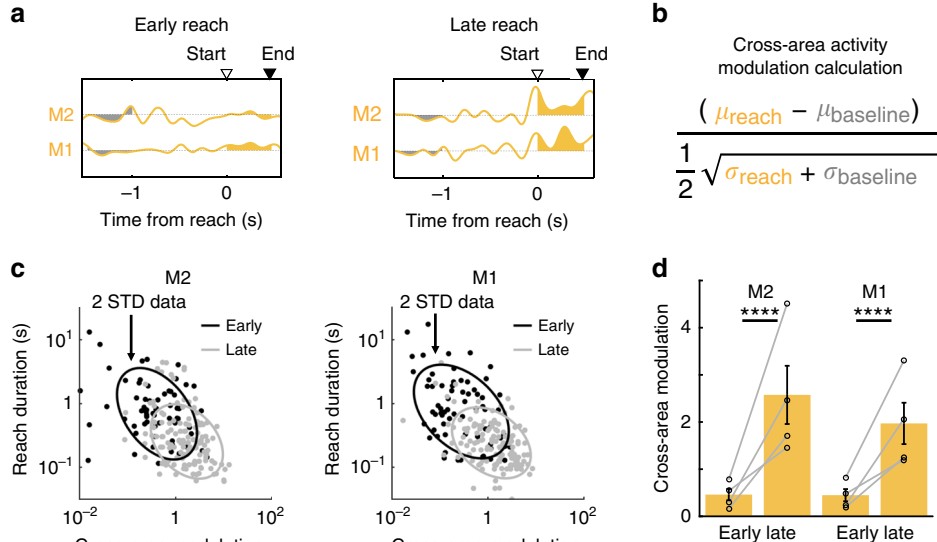

**Fig. 5 Learning amplifies cross-area representations of reach duration. a** Example single-trial M2 and M1 cross-area activity in early (left) and late (right) learning. Reach duration is indicated by triangles marking reach start (open triangle) and reach end (filled triangle). Single-trial baseline modulation in gray, reach modulation in yellow. **b** Equation for calculating cross-area activity (CA) modulation (see Methods). **c** Single-trial CA-modulation predicts single-trial reach duration. Single-trial CA-modulation for M2 (left) and M1 (right) cross-area activity is plotted against single-trial reach duration. Points show randomly subselected trials from early (black) and late learning (gray), with ellipses fitted to 2 standard deviations of the full datasets. All trials were used for quantification. **d** CA-modulation increases in both M1 and M2 with learning. Gray lines, also marked by black open circles, show values for individual animals, bars show mean ± std. dev., $n = 4$ rats. ****$p < 0.0001$. one-sided hierarchical bootstraps, not adjusted for multiple comparisons.

coordinating network-wide activity related to salient behaviors during learning.

**M2 local activity precedes M1 local activity**. A prominent model of M2–M1 interactions during learning proposes a strong top–down influence from M2 to M1[3,4]. While cross-area activity may coordinate areas, we expected that local activity in each region might reflect a M2-to-M1 top–down relationship. For example, if M2-specific local activity temporally preceded M1-specific local activity, it would suggest that M2 is more likely to play a top–down role. To address this, we first defined local activity as the population firing not accounted for in cross-area activity. For each 100 ms time bin, we projected the population firing rate vector onto the hyperplane orthogonal to the CCA subspace, and used the magnitude of this vector as our local activity value (Fig. 6a). We then quantified, on a single-trial basis, the difference in median timing of M2 and M1 local activity in early and late learning (Fig. 6b, c, see Methods). We found that M2 local activity consistently preceded M1 local activity in both early and late learning (Fig. 6d). These timing differences were significant when quantified via permutation testing. Specifically, we randomly assigned trial activity to either M1 or M2, computed the timing differences of the permuted data ($10^5$ permutations), and used these differences as a reference distribution to evaluate the significance of the M2–M1 timing differences ($p < 0.00001$) (Fig. 6e). We used a similar approach to evaluate the change in M2–M1 temporal coupling. We thus randomly assigned M2–M1 timing differences to either early or late learning, computed the means of the permuted datasets ($10^5$ permutations), and used the difference of those means as a reference distribution to evaluate the significance of the M2–M1 temporal coupling change with learning. M2 and M1 local activity became more tightly coupled with learning, as quantified by the significant decrease in the timing gap between their local activities with learning ($p < 0.00001$) (Fig. 6f). These timing results are consistent with a M2 to M1 top–down hierarchical relationship.

**M2 inhibition disrupts skilled reaching**. Based on our results, we expected M2 activity to be necessary for improvements in behavior with learning, as well as for amplified representations of learned movements in M1 cross-area activity. If activity shared between M2 and M1 helped to shape M1 representations, then disrupting M2–M1 cross-area activity should impact reaching behavior. To test this, we inactivated M2 in well-trained animals using the GABA agonist muscimol (Fig. 7a). Unlike control saline infusions (Supplementary Fig. 5), M2 inactivation caused severe performance deficits, with reaching behavior qualitatively similar to early learning (Fig. 7b). M2 inactivation decreased success rate (hierarchical bootstrap, 56.75% ± 5.16 for Baseline, 37.45% ± 6.88 for Muscimol, $p = 0.0082$, $n = 6$ rats), increased reaction time (hierarchical bootstrap, 1.26 s ± 0.28 for Baseline, 3.23 s ± 0.74 for Muscimol, $p < 0.0001$, $n = 6$ rats), and increased reach duration (hierarchical bootstrap, 0.18 s ± 0.018 for Baseline, 0.29 s ± 0.035 for Muscimol, $p < 0.0001$, $n = 6$ rats). M2 saline did not decrease success rate (hierarchical bootstrap, 54.81% ± 5.40 for Baseline, 57.84% ± 4.18 for Saline, $p = 0.7453$, $n = 6$ rats), but did cause small but significant increases in reaction time (hierarchical bootstrap, 1.60 s ± 0.37 for Baseline, 2.13 s ± 0.49 for Saline, $p = 0.0021$, $n = 6$ rats) and reach duration (hierarchical bootstrap, 0.20 s ± 0.024 for Baseline, 0.22 s ± 0.025 for Saline, $p = 0.0452$, $n = 6$ rats).

**M2 inhibition preferentially disrupts M1 cross-area activity**. To examine the influence of M2 inactivation on M1 neural representations, we performed simultaneous recordings in M1 and M2 during baseline performance and during M2 inactivation on the same day in well-trained animals. This approach allowed us to define the M2–M1 cross-area activity space with M2 intact, then track the effects of M2 disruption on single-unit M1 activity, M1 cross-area dynamics, and M1 local dynamics. First, we compared movement modulation of M1 single neurons during baseline reaches and M2 inactivation. Remarkably, not only did M2 inactivation disrupt M1 single-neuron movement modulation (hierarchical bootstrap, 48.43% ± 16.93 for Baseline, 24.88% ±

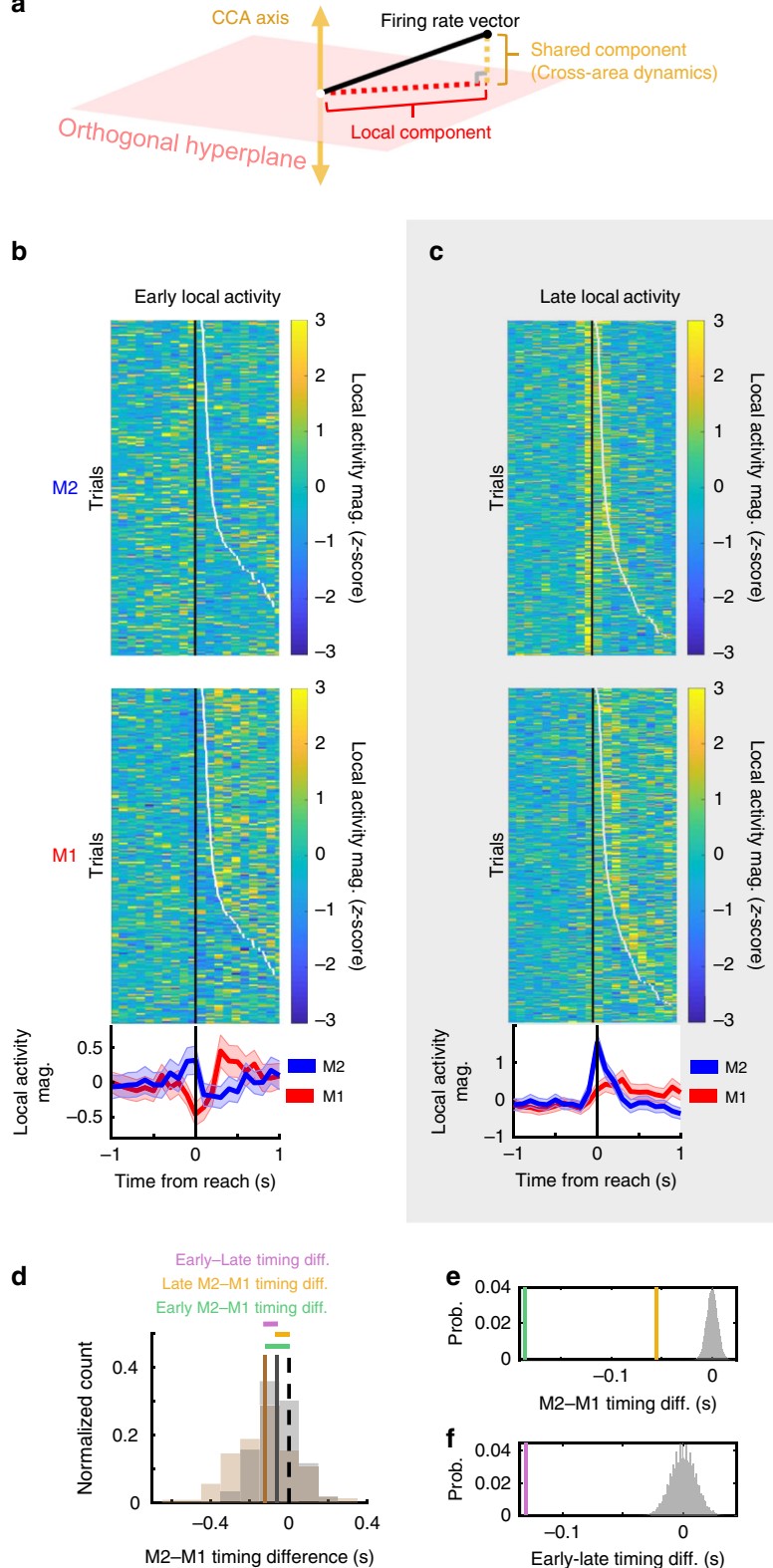

17.01 for M2 Muscimol, $p = 0.0137$, $n = 3$ rats) (Fig. 7c), but the M1 neurons which contributed most to M2–M1 cross-area activity (i.e., with the highest magnitude CCA weights) experienced greater drops in movement modulation (linear regression on log(CCA weight mag.), $R^2 = 0.12$, $t = 2.81$, $p = 0.0067$, $n = 60$ neurons from three rats) (Fig. 7e). This result is consistent with

the model that M2 population activity shapes M1 population activity through M2–M1 cross-area signals.

Our data predicted that M2 inactivation might preferentially affect M1–M2 cross-area population dynamics, thereby removing top–down influence on M1, with minimal disruption of local M1 computations. Intriguingly, we found that M2 inactivation

**Fig. 6 Local signals support a M2 to M1 hierarchy. a** Illustration of approach to identifying local activity using artificial data. Black line represents a population firing rate for a single time bin. Population firing was projected onto a shared axis defined by CCA (solid yellow line) to obtain the cross-area signal (dotted yellow line), and onto the hyperplane orthogonal to the CCA subspace (light red plane), to obtain the local signal (dotted red line). **b** Example animal local activity in early learning (z-scored). (Top) Single-trial local activity trajectories for M2. Black dots indicate reach onset and white dots indicate transition to grasp on each trial. (Middle) Single-trial local activity trajectories for M1. (Bottom) Mean local activity trajectories for M2 (blue) and M1 (red). Shaded regions show SEM, $n = 211$ trials. **c** As in **b**, but for late learning, $n = 297$ trials. **d** Distributions of timing differences between median timing of single-trial M2 and M1 local activity in early (brown) and late learning (gray) (from example animal in **b**, **c**). Black dotted line indicates zero lag in M2–M1 median timing of local activity. **e** Quantification of M2–M1 timing differences ($n = 4$ rats). In gray, permutation-based reference distribution of timing differences with data randomly assigned to M1 or M2 ($10^5$ permutations). M2 local activity significantly preceded M1 activity in early (green) and late (orange) learning. **f** Quantification of tighter coupling between M2 and M1 local activity from early to late learning ($n = 4$ rats). In gray, permutation-based reference distribution of mean difference in M2–M1 timing coupling with M2–M1 timing differences randomly assigned to early or late learning ($10^5$ permutations). The true difference in M2–M1 coupling between early and late learning (purple line) was more negative than any of the reference values, indicating that the timelag between M2 local activity and M1 local activity significantly decreased with learning.

disrupted movement modulation of M2–M1 cross-area activity significantly more than the movement modulation of local M1 activity (Fig. 8a). We quantified this by computing single-trial peak-to-trough values on a 1-s interval centered on reach start for both cross-area and local signals. We found that across animals, both cross-area and local signals had decreased modulation with M2 inactivation, but that the decrease was significantly greater for cross-area signals (hierarchical bootstrap; M2 muscimol resulted in a 58.19% ± 8.71 decrease in peak-to-trough amplitude for M1 cross-area signals, and a 26.52% ± 7.20 decrease for M1 local signals, $p = 0.0042$, $n = 3$ rats). These results indicate a degree of independence between cross-area and local dynamics. This decoupling may provide a mechanism for resilience of local dynamics, improving robustness in the event of distant network damage.

M2 inactivation resulted in a small but significant drop in firing rate across the M1 population (hierarchical bootstrap, 24.65 Hz ± 2.73 for Baseline, 19.15 Hz ± 4.64 for M2 Muscimol, $p < 0.0001$, $n = 3$ rats). However, histological analysis confirmed that this was not a direct effect of the muscimol itself, which did not reach M1 (Supplementary Fig. 6). Instead, this effect may be due to loss of M2 inputs[49]. Furthermore, mean M1 local covariance did not change, indicating stability in local M1 functional connectivity (two-sided hierarchical bootstrap, 0.24 ± 0.063 shared variance/total variance for Baseline; 0.19 ± 0.043 shared variance/total variance for M2 Muscimol, $p = 0.28$, $n = 3$ rats, see Methods).

**M2 inhibition disrupts M1 representation of movement**. We found that M2 inactivation decreased M1 cross-area modulation during reach initiation (Fig. 8b–e). We quantified this by comparing the difference in median M1 activity along the cross-area activity axis before reach and at reach initiation during baseline trials and M2 inhibition trials (Fig. 8b–d). We found that the difference between M1 cross-area neural activity during pre-reach and reach initiation was significantly smaller during M2 inhibition (mixed-effect model, 0.35 ± 0.06 for Baseline, 0.03 ± 0.09 for M2 Muscimol, $t(4) = -3.57$, $p = 0.02$, $n = 3$ rats). As before, we fit a logistic regression model to predict reach onset from M1 cross-area activity. We quantified the model's performance and saw that detection of reach initiation based on M1 cross-area activity decreased with M2 inhibition (mixed-effect model, 0.64 ± 0.03 for Baseline, 0.52 ± 0.04 for M2 Muscimol, $t(4) = -3.48$, $p = 0.02$, $n = 3$ rats) (Fig. 8e), indicating that M1 cross-area dynamics were less informative about reach initiation during M2 inhibition. The loss of information about reach initiation caused by M2 inactivation can be directly contrasted to the learning-driven gains in information about reach initiation that are illustrated in Fig. 4.

In addition to disrupting reach initiation signals, we also found that M2 inhibition decreased reach modulation of M1 cross-area dynamics (hierarchical bootstrap, 0.78 ± 0.21 for Baseline, 0.27 ± 0.050 for M2 Musimol, $p < 0.0001$, $n = 3$ rats) (Fig. 8f). This indicated that M2 input is necessary for intact M1 reach modulation and implied a M2 to M1 directionality. We additionally examined whether M2 inactivation entirely disso- ciated M1 cross-area modulation from behavioral performance. We found that the relationship between reach duration and M1 CA-modulation was still significant during M2 inactivation (mixed-effect model, log slope = −0.26, $t(187) = -5.54$, $p = 9.99 \times 10^{-8}$, $n = 3$ rats) (Fig. 8g), underscoring the fundamental relationship between M1 and behavior even during motor system disruption.

## Discussion

This study outlines a new approach to understanding simulta- neous activity shared between two cortical areas. First, we demonstrate that a computational method identifying maximally correlated activity patterns between regions can be used to isolate cross-area population dynamics. Second, we show that cross-area population dynamics become more related to both reach initia- tion and duration with learning. Through causal manipulations, we found that local M2 inactivation disrupted M1 cross-area dynamics as well as skilled reach execution. The M1 activity remaining in the M1–M2 cross-area dynamics axes was still predictive of single-trial behavior, indicating maintenance of meaningful movement activity in M1. However, M2 muscimol inactivation led to slower reactions to environmental cues and less efficient reaches, consistent with the hypothesis that attenuation of M2–M1 cross-area activity impairs M2 top–down guidance of behavior. These results demonstrate that M1–M2 cross-area dynamics represent and contribute to skilled execution.

There are two common approaches to understanding M2 and M1 signals during movement. First is to compare neural signals from each region in order to detect differences supporting their distinct roles. This approach has led to a model of M2 and M1 functioning within a hierarchy, with M2 providing top–down signals to M1 related to movement planning[4,5,16], timing[50], and context[3,50,51]. In contrast, the second approach focuses on simi- larities between M1 and M2. Because M1 and M2 are both highly active during well-learned movements, many studies combine activity from both regions to understand cortical single-neuron and population dynamics during movement[24,28,52]. This allows analysis of a larger number of neurons with similar relationships to movement. These two approaches emphasizing the differences and similarities in M2 and M1 data are not mutually exclusive, but they are rarely[5] combined to understand how cross-area versus local signals contribute to network function.

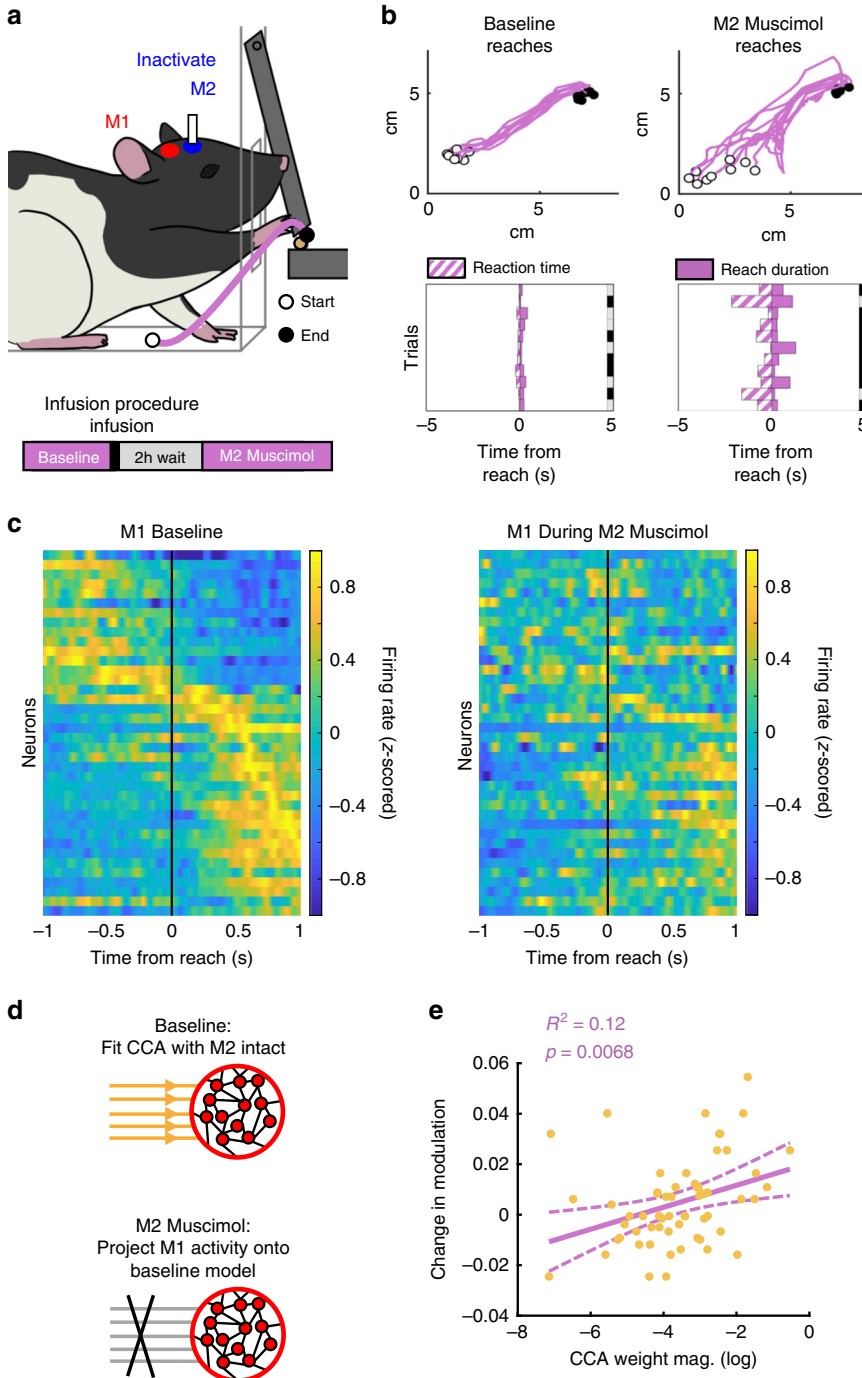

**Fig. 7 M2 inhibition disrupts learned reach behavior and M1 single-unit movement modulation. a** Rats previously trained on the reach-to-grasp task were infused with muscimol in M2. (Bottom) Experimental paradigm for evaluation of reach behavior during M2 inactivation (see Methods). **b** (Top) Example reach trajectories during baseline (left) and M2 muscimol (right) trials. White dot marks reach start position. Black dot marks reach end position. (Bottom) Example consecutive single-trial representations of reaction time (purple striped bars) and reach duration (purple bars) for baseline (left) and M2 muscimol inactivation (right). Right border of plot shows accuracy, with success in gray and failure in black. **c** Population PETH for M1 units across $n = 3$ rats during baseline (left) and muscimol inactivation (right). Each neuron's PETH was tested for trial modulation during the baseline period using a circular shuffle test. Only neurons that were significantly modulated during the baseline period are shown, ordered by their peak time during the baseline period in both plots. Firing rates are z-scored per-neuron. **d** Due to the same-day inactivation paradigm, CCA weights could be computed for the baseline session, and then used with the M1 data recorded during the M2 muscimol session. **e** Across animals, for each neuron (yellow dot), the change in task modulation between baseline and M2 inactivation was computed (baseline − M2 muscimol), and compared with that neuron's absolute value CCA weight from the baseline period. The regression fit (purple) shows that neurons with higher magnitude CCA weights tended to have larger drops in modulation. Two-sided t-statistic, not adjusted for multiple comparisons.

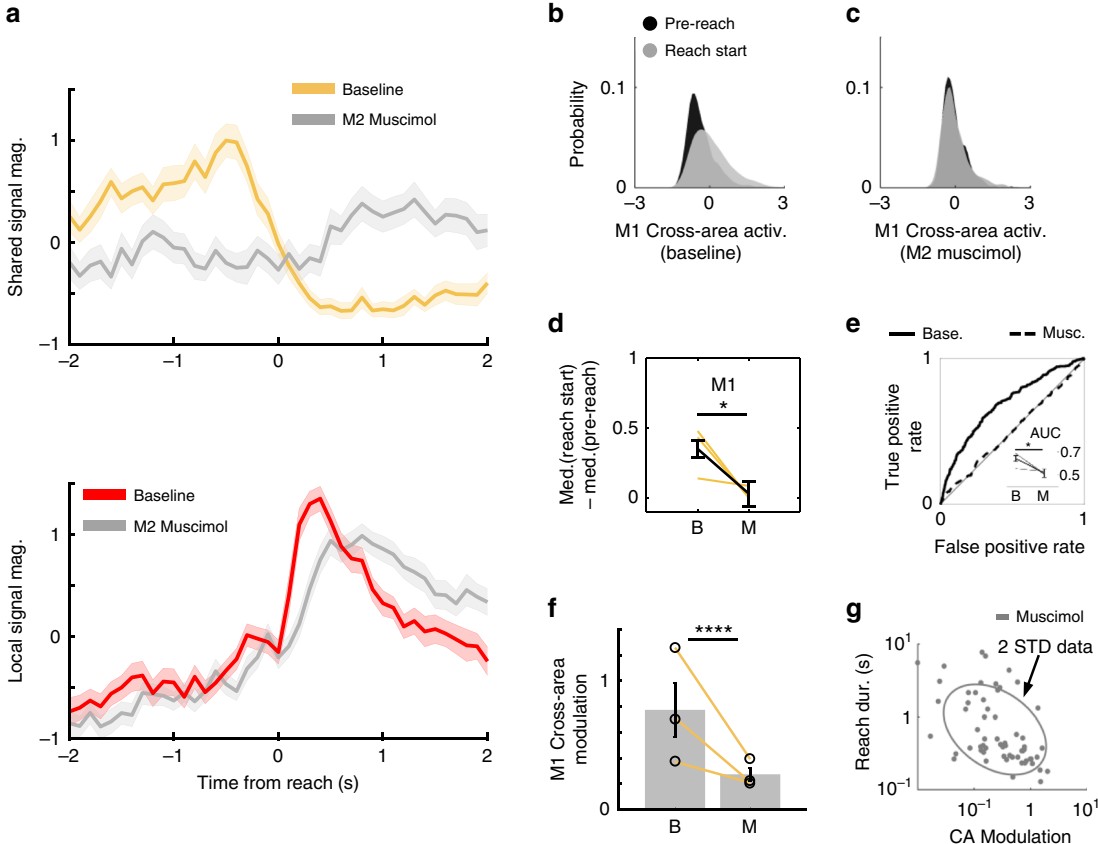

**Fig. 8 M2 inhibition disrupts M1 cross-area movement modulation. a** (Top) Comparison of z-scored, trial-averaged M1 cross-area activity magnitudes during baseline (yellow) and M2 inactivation (gray) for one example animal. M1 CCA weights were defined during baseline period and used to calculate cross-area activity during both baseline and M2 inactivation trials. Solid line shows mean and shaded region shows standard error of the mean. (Bottom) As above, but for M1 local signals during baseline (red) and M2 inactivation (gray), suggesting that local signals are not as impacted by M2 inactivation. n = 91 trials for Baseline, n = 92 trials for M2 Muscimol. **b** Probability density function of M1 cross-area neural activity during baseline trials. Pre-reach activity in black. Reach start activity in gray. **c** As in **b**, but during M2 inactivation trials. **d** Quantification of (**b**, **c**) as the difference between median pre-reach and reach activity during baseline and M2 inactivation trials in M1 cross-area subspace. Yellow lines show data from individual animals, black line shows mean ± SEM, n = 3 rats. p = 0.02. *p < 0.05. two-sided t-statistic, not adjusted for multiple comparisons. **e** Detection of reach initiation from M1 cross-area subspace activity using ROC analysis on logistic regression model (example animal). (Inset) Difference in reach detection quantified as the area under the curve (AOC) for all animals. Gray lines show data from individual animals; black lines show mean ± SEM, n = 3 rats. p = 0.02. *p < 0.05. two-sided t-statistic, not adjusted for multiple comparisons. **f** M1 cross-area activity (CA) modulation decreases significantly with M2 inactivation. Yellow lines, also marked with black open circles, show data from individual animals, bars show mean ± std dev., n = 3 rats. ****p < 0.0001. One-sided hierarchical bootstrap, not adjusted for multiple comparisons. **g** Single-trial M1 CA-modulation predicts single-trial reach duration even during M2 inactivation. Plot shows random subsampling of trials across animals, all trials were used in quantification.

Here we analyzed both the activity simultaneously shared across M1 and M2, and the otherwise unaccounted-for activity in each region. In any brain region, exclusively local activity is impossible to identify in-vivo, as this would require comprehensive recordings from the entire brain in order to account for activity shared with any other region. However, M1 and M2 have heavily overlapping inputs and outputs[53], allowing us to consider the M1–M2 cross-area activity as encompassing most of each region's brain-wide shared activity. We found that the local dynamics in M2 and M1 had a clear temporal relationship, with M2 preceding M1, consistent with the pervasive model of top–down M2-to-M1 signals[3,4]. While this analysis quantified trial-by-trial timing in M2 and M1 local dynamics during early and late learning, it remains unknown whether more detailed analysis of M2 and M1 local dynamics would reveal additional trial-by-trial processes for top–down learning. However, as learning signals may initially be unpredictably related to many aspects of motor learning (timing, vigor, etc.) such work may be better accomplished in a

dataset with more neurons from animals performing simpler behaviors.

Shared neural population dynamics have been identified within single brain regions[7,11,28], between two hierarchical cortical regions[5,16,32], and across sets of functionally diverse brain regions[17,18,33]. In many cases, shared dynamics are defined solely on functional relationships in neural activity, independent of behavior (but see ref. [54]). This approach moves away from the view of fluctuations in neural signals as noise in a stochastic system. Instead, it frames neural activity as encompassing a range of neural computations without apparent relation to behavior[18,24,28,55]. Despite this behaviorally independent approach to understanding neural signals, one common thread in studies of cross-area activity is the dominance of movement signals within shared dynamics[18,46]. This privileged representation of movement emphasizes its importance as the final output of the nervous system, and suggests the possibility that movement signals have a role in shaping neural activity in a broad array of functional systems, including cognitive and motivational circuits.

Here, we use learning as an intervention to change the behavioral salience of reach-to-grasp movements. In other words, we probe how learned behavioral salience of a given movement changes its representation in cross-area neural signals. We found that cross-area representations of similar movements were less modulated in early learning than late learning, consistent with a functional role for cross-area activity in amplifying neural signals for salient behaviors. This interpretation is consistent with a study of correlated variability between neurons in V4[48], which found that the relationship between correlated variability and performance was the same for performance improvements driven by either attention or learning, two manipulations of behavioral salience. Thus, we posit that shared activity specifically modulates neural representations for salient behaviors.

Past work has proposed that the role of M2 is to provide top–down control and contextual information to M1[4,51]. Here, we provide insight into what such a signal might look like, and how it evolves with learning. In early learning, when behavior was exploratory and variable, high amplitude cross-area dynamics were less related to specific behavioral timepoints, and modulation of cross-area activity was weakly related to reaching. However, even at this early stage, reaches with more movement-modulated cross-area activity tended to have shorter durations. After learning, the relationship between cross-area activity modulation and behavior was amplified. Notably, the single-trial M1–M2 cross-area dynamics corresponding to similarly efficient, short duration reaches were not identical in early and late learning. This argues against the notion that pre-existing representations of efficient movements are selected through learning[56]. Instead, our results support the idea that learning transforms and amplifies the neural signals for behaviors that are selected[10]. This finding also highlights the feasibility and importance of analyzing single-trials in order to understand highly variable behavioral states such as early learning.

Finally, the influence of cross-area dynamics on behavior appears to be causal, since M2 inactivation disrupted both M1 cross-area dynamics and reaching behavior, while local properties of M1 were less affected. Examining local activity during upstream inactivation provides a valuable approach to differentiating between activity dynamics generated locally and those from top–down influences. Such analyses are impossible in purely correlative studies, and, paired with same-day establishment of cross-area dynamics, demonstrate a novel approach to understanding how several patterns of covariance and information encoding overlap[16,18,32,54] and interact within functional neural systems[5,32]. Furthermore, we found that when M2 inputs are removed, M1 local shared variance does not change. This is important because there has been increasing concern that acute changes in input to an area can perturb behaviorally relevant local population relationships[49,57]. Importantly, rats do produce some successful reaches during both early learning and M2 inactivation, although they are less frequent and less efficient. This demonstrates that M1 can independently produce functional reach-to-grasp behavior, and suggests that top–down input from M2 is a learned signal, biasing M1 towards more effective behavior. This is concordant with long-standing models of top–down M1–M2 interactions during learning[3] and reinforces the view that, while M2 and M1 both represent movement, M2 is particularly important for learned, complex skills[2,4,6,51,58–60].

Our results provide direct evidence that M1–M2 cross-area dynamics reflect task learning and single-trial skill performance. Knowledge of this phenomenon should help to better understand mechanisms of neural plasticity and functional properties of large-scale, hierarchical networks in the context of flexible learned behaviors.

## Methods

**Animal care.** All procedures were in accordance with protocols approved by the Institutional Animal Care and Use Committee at the San Francisco Veterans Affairs Medical Center. Adult male Long–Evans rats ($n = 10$, 250–400 g; Charles River Laboratories) were housed in a 12-h/12-h light–dark cycle. All experiments were done during the light cycles. Rats were housed in groups of two animals prior to surgery and individually after surgery.

**Surgery.** All surgical procedures were performed using a sterile technique under 2–4% isoflurane. Surgery involved cleaning and exposure of the skull, preparation of the skull surface (using cyanoacrylate) and then implantation of the skull screws for overall headstage stability. Reference screws were implanted posterior to lambda and ipsilateral to the neural recordings. For experiments involving physiological recordings, craniotomy and durectomy were performed, followed by implantation of the neural probes. For experiments involving only infusions, burr holes were drilled in the appropriate locations, followed by implantation of the cannulas. Postoperative recovery regimen included the administration of 0.02 mg kg$^{-1}$ buprenorphine for 2 days, and 0.2 mg kg$^{-1}$ meloxicam, 0.5 mg kg$^{-1}$ dexamethasone and 15 mg kg$^{-1}$ trimethoprim sulfadiazine for 5 days. All animals were allowed to recover for 1 week prior to further behavioral training.

**Electrode array and cannula implants.** Long–Evans hooded rats were implanted with two 32-channel tungsten wire probes (TDT or Innovative Neurophysiology), one each in M1 (+0.5 AP, +3.5ML, −1.5 DV)[27,38,44,61] and M2 (+4.0–4.5 AP, +1.5ML, −1.5 DV)[61,62], contralateral to reaching arm (see Supplementary Table 2). Infusion cannulas were implanted in M2 (+4.0 AP, +1.5ML, −1.5 DV) for infusion-only animals. For rats with both M2 electrode arrays and cannulas, the cannula was attached to the lateral side of the electrode array prior to surgery.

**Functional ICMS mapping.** Two additional animals were used to confirm that forelimb movement could be evoked from both M1 and M2 (see Supplementary Fig. 1). For the two mapping procedures, animals were initially anesthetized with a mixture of ketamine hydrochloride (100 mg kg$^{-1}$) and xylazine (16.67 mg kg$^{-1}$) delivered intraperitoneally. Supplementary 0.5–1 cc doses of the mixture were provided as needed, based on toe-pinch response. 32-channel tungsten microwire electrode arrays (Tucker Davis Technologies, ~50 kΩ input impedance at 1000 Hz) were implanted in M1 ($n = 1$) and M2 ($n = 1$), at a depth of 1500 μm, targeting cortical layer V.

Consistent with prior studies[63,64], triplet biphasic trains of 200 μs per phase (100 μs inter-phase interval, 333 Hz triplet) were delivered at each electrode using a constant current stimulator (IZ2, TDT) controlled by a custom Synapse program (TDT). These trains were delivered with 60–150 μA amplitude[64,65]. Movements were evoked across large portions of the M1 and M2 arrays (Supplementary Fig. 1). Animals were placed in a prone position such that the contralateral forelimb remained free. Stimulation was delivered at each electrode in the array with video recording at 20 frames per second. Movement, if elicited, was visible immediately after onset of stimulation, with greater amplitude of movement at higher currents and frequencies.

**Pharmacological infusions.** Rats were anesthetized with 2% isoflurane before infusions. We injected 0.5–1 μL (1 μg μL$^{-1}$)[44,60] of the GABA receptor agonist muscimol into contralateral M2 (infusion rate: 1 nL min$^{-1}$) through a chronically implanted cannula using a Hamilton infusion syringe. The infused volumes were titrated for each animal. We first started with the larger volume (1 μL). If the animal was unable to reach within 2 h, we downscaled to the smaller volume (0.5 μL). The infusion syringe was left in place for at least 5 min post infusion. Rats were allowed to recover in their home cages for 2 h before starting behavioral testing.

**Histology.** Final placement of the electrodes was monitored online based on implantation depth and verified histologically at the end of the experiments. Rats were anesthetized with isoflurane and transcardially perfused with 0.9% sodium chloride, followed by 4% formaldehyde. The harvested brains were postfixed for 24 h and immersed in 20% sucrose for 2 days. Coronal cryostat sections (40-μm thickness) were mounted with permount solution (Fisher Scientific) on superfrosted coated slides (Fisher Scientific). Images of a whole section were taken by a HP scanner, and microscope images were taken by a Zeiss microscope.

**Behavioral training.** We used an automated behavior paradigm to train rats to perform dexterous reach-to-grasp movements[36]. Rats learn to reach through a narrow slot to grasp and retrieve a 45 mg pellet from a shallow dish (i.e. pellet holder) placed ~1.5 cm outside the behavioral box[35]. Prior to implantation, rats were handled and habituated to the behavioral box for at least 1 day, then manually prompted to reach for a pellet 10–30 times to determine handedness. Handedness was determined when rats reached with the same hand for $> = 70\%$ of at least ten test trials. The start of each trial was signaled with a tone and the opening of a door allowing access to the pellet. Trials ended when the door was closed, which was triggered either by the pellet being dislodged from the pellet holder, or, if this did not occur, ~15 s after door opening.

**Behavioral training for learning animals.** Once handedness was determined, rats were implanted with neural probes (see Surgery). For 2 days before behavioral training, rats were food restricted, followed by feeding animals a fixed amount during the course of training. During behavioral training, rats were placed in an automated reach box and completed 38–300 trials per day. The early learning training day was the first day on which the rat completed at least 30 trials. The late learning training day was the second consecutive day on which the rat performed with at least 45% success rate (see Supplementary Table 1).

**Behavioral training for M2 inactivation animals.** Once handedness was determined, rats were trained until their success rate reached a plateau (at least 2 consecutive days with performance above 45% and >100 completed trials/day), after which they were implanted with infusion cannulas alone ($n = 3$ rats), or with infusion cannulas and electrodes ($n = 3$ rats) (see Surgery). Rats were allowed at least a week of recovery after surgery before beginning behavioral testing. Rats were retrained until plateau performance (>2 consecutive days with performance above 40%). On M2 inactivation days, rats performed ~100 reach trials before receiving pharmacological infusions. After 2 h of rest post infusion, rats were retested for ~100 trials (see Supplementary Table 1).

**Behavioral analysis.** Rat behavior was video recorded using a side view camera (30–100 Hz, see Supplementary Table 1) positioned outside the behavioral box, perpendicular to the main direction of movement. Each rat's reach hand was painted with an orange marker at the start of each day. Reach videos were viewed and semi-automatically scored to obtain trial success, hand position, timepoints for reach onset, and grasp onset. To characterize motor performance, we quantified reach duration, reaction time, maximum movement speed, and pellet retrieval success for each trial. Percent reach success is the percent of trials on which the pellet was retrieved during a single day of training, excluding trials in which the rat did not dislodge the pellet from the holder or displayed abnormal behavior (i.e., licking, reaching with the wrong hand). Reach duration for each trial was defined as the time from the start of reach to onset of grasping or when the paw first touched the pellet if no grasping occurred on that trial. Reaction time was defined as the time between the door open cue and movement onset—note that since the rat was freely moving in the behavior box, reaction time is affected by both the rat's position and attention at the time of the cue.

**Electrophysiology data collection.** We recorded extracellular neural activity using tungsten microwire electrode arrays (MEAs, $n = 8$ rats, TDT or Innovative Neurophysiology). We recorded spike and LFP activity using a 128-channel TDT–RZ2 system (TDT). Spike data was sampled at 24,414 Hz and LFP data at 1018 Hz. Analog headstages with a unity gain and high impedance (~1 GΩ) were used. Snippets of data that crossed a high signal-to noise threshold (4 standard deviations away from the mean) were time-stamped as events, and waveforms for each event were peak aligned. MEA recordings were sorted offline using either superparamagnetic clustering program (WaveClus[66]) or a density-based clustering algorithm (Mountainsort[67]). Clusters interpreted to be noise were discarded, but multi-units were kept for analysis. Trial-related timestamps (i.e., trial onset, trial completion, removal of pellet from pellet holder, and timing of video frames) were sent to the RZ2 analog input channel using an Arduino digital board and synchronized to neural data.

**Cross-area neural subspace and population dynamics.** Shared cross-area subspaces (CSs) were defined using CCA, which identifies maximally correlated linear combinations between two groups of variables. Neural data in M2 and M1 was binned at 100 ms, and data from −1 to +1 s surrounding time of grasp onset was concatenated across trials and mean subtracted. CCA models were fit using the MATLAB function *canoncorr*. For analyses and figures involving times outside the −1 to +1 s window around grasp, data from other time periods was projected onto these models.

CCA produces as many CVs as the number of neurons in the smaller population (e.g., if there are 30 M2 neurons and 20 M1 neurons, CCA will fit 20 CVs). The $R^2$ values of each CV were computed using tenfold cross-validation, and the $R^2$ values reported in Fig. 3b are for the top CV only. The cross-validation procedure used to compute the $R^2$ values is as follows: The full dataset was randomly partitioned into ten equal folds (ignoring trial structure, i.e. timepoints from the same trial could be assigned to different folds). Then, ten different times, one fold was assigned to be the test data and the other nine to be the training data. CCA models were fit to the training dataset. The test data was then projected onto the training model, and $R^2$ values were computed between the M1 and M2 projections for each CV. The $R^2$ values reported for each CV are the average across all ten combinations of testing/training data, and are intended to measure how well the models generalize to held-out data. Other than when reporting $R^2$ values (Figs. 2d, 3b; Supplementary Fig. 3a, d), or comparing weights fit on different subsets of data (Fig. 2c), the CCA models used were those fit to the full datasets.

To determine which CVs were significant, the $R^2$ of each CV was compared with a bootstrap distribution made of the $R^2$ of the top CV from CCA models fit to trial-shuffled data ($10^4$ shuffles). Specifically, before fitting CCA, trials from M2 were concatenated in the order in which they occurred, while trials from M1 were

randomly permuted prior to concatenation. This method maintains local neural patterns, as well as neural modulation which could be attributed to coarse behavioral variables that do not vary by trial, while breaking moment-by-moment relationships between the regions. Therefore, computing CCA on trial-shuffled data provides a floor for the degree of correlation expected from the fact that many neurons in both regions have firing rate fluctuations around the time of grasp. A CV was considered significant if its $R^2$ was greater than the 95th percentile of the bootstrap distribution. One animal was eliminated from further analysis because its early dataset had no significant CVs. All other datasets had 1–3 significant CVs. For evaluating cross-area signals (Figs. 2–5, 7e, 8b–g; Supplementary Figs. 3a–c, 4), only the top CV was used, as this provided a consistent dimensionality across datasets, and a signal with both magnitude and sign.

To test whether results were unique to the 100 ms bin width, CCA models were also fit to data binned at 75 and 50 ms (Supplementary Fig. 3). Qualitatively, CCA trajectories fit to smaller binwidths appeared noisier but had peaks at similar timepoints as the 100 ms models. There was no significant difference in $R^2$ between the 100 ms bin width and the 75 ms bin width, but the 50 ms bin width had a significantly smaller $R^2$ than either the 100 or 75 ms models (two-sided hierarchical bootstrap, $10^4$ shuffles; $0.27 \pm 0.047$ for 100 ms models, $0.28 \pm 0.046$ for 75 ms models, $0.22 \pm 0.037$ for 50 ms models; $p = 0.83$ for 100 ms vs 75 ms; $p = 0.016$ for 100 vs 50 ms; $p < 0.0002$ for 75 vs 50 ms; $n = 160$ $R^2$ values from eight datasets each with tenfold cross-validation for each condition), suggesting that larger bin sizes are needed to capture the cross-area signal. We also compared the angle between the top CV for models fit to 100, 75, and 50 ms data (hierarchical bootstrap, $10^4$ shuffles; $13.74° \pm 2.40$ for angle between 100 ms model weights and 75 ms model weights, $20.04° \pm 3.05$ for 100 vs 50 ms, $18.22° \pm 2.62$ for 75 vs 50 ms; $n = 8$ models per bin width, each fit to early or late data from four rats). In M2, we found that the angle between the 100 and 75 ms models was significantly smaller than both the angle between the 100 and 50 ms models ($p = 0.0094$) and the angle between the 75 and 50 ms models ($p = 0.0041$). In M1, we found that the angle between the 100 and 75 ms models was significantly smaller than the angle between the 100 and 50 ms models ($p = 0.00089$) but not significantly smaller than the angle between the 75 and 50 ms models ($p = 0.097$). For all models, all bootstrap samples were less than 45°, suggesting that models fit to different binwidths identified similar patterns of covariation. In addition, we fit CCA models for all three binwidths at timelags from −500 to +500 ms, and found that for six of eight datasets, using 100 ms bins with no lag resulted in the highest $R^2$ values.

**Normalized cross-correlation.** Normalized cross-correlations were calculated as the peak correlation coefficient for timelags between −200 and +200 ms minus the mean correlation coefficient for all timelags in that range.

**Local neural subspace and population dynamics.** Local signals were computed by projection onto the hyperplane orthogonal to the CS defined by CCA. For comparison with local signals (Figs. 6, 8a), all significant CVs were used to define the CS, so that the local signal would be orthogonal to all significantly correlated cross-area activity. This meant that dimensionality varied across datasets, and the signals analyzed were the magnitudes of the projections onto the cross-area and local subspaces.

**Reach start decoding.** To calculate the difference in CS activity before reach initiation versus during reach initiation, we defined a pre-reach period as −2 to −0.1 s before reach initiation and a reach initiation period from −0.1 to +0.3 s surrounding reach initiation. CS activity from each of these periods was concatenated across trials to then calculate the median CS activity value. The difference between median CS activity during pre-reach and reach initiation was calculated for each animal.

For reach start prediction, activity from pre-reach and reach initiation was labeled as 0 or 1, respectively, which was then used as the response values to train a logistic regression model using the MATLAB function *fitglm*. The probability that CS activity values corresponded to a timepoint during reach initiation was returned as scores. We then used these scores to compute the receiver operating characteristic (ROC) curve of the classification results using the MATLAB function *perfcurve*. The area under the curve (AUC) was returned for each animal, and these values were used in mixed-effect modeling to detect difference in pre-reach versus reach initiation activity during early versus late learning, and baseline versus muscimol behavior.

The logistic regression model was used to calculate the probability of reach initiation based on CS activity on single trials. We calculate the single-trial difference in the mean predicted probability of reach initiation during the pre-reach versus reach initiation periods. We compared this difference using all trials in early versus late learning, and baseline versus muscimol behavior. We plotted the median probability of reach initiation across trials aligned to reach initiation.

**Neural reach modulation.** Single-trial neural reach modulation of the first CV from CCA was calculated using the signal processing $d'$ (d-prime) signal sensitivity metric defined by the equation below[68], where $\mu$ indicates the mean and $\sigma$ indicates the standard deviation of the signal. For each trial, the reach period was defined as −0.1 s before reach onset to +0.1 s after grasp onset; the baseline period was

defined as a length of time equal to the reach period, ending 1 s before the start of the reach period. To produce single-trial normalization, the median value from the baseline period was subtracted from both the movement signal and the baseline signal before calculating the single-trial modulation value ($d'$), as below.

$$d' = \frac{\mu_{\text{reach}} - \mu_{\text{baseline}}}{\frac{1}{2}\sqrt{\sigma_{\text{reach}} + \sigma_{\text{baseline}}}}. \qquad (1)$$

**Mean local covariance**. Each neuron's shared over total variance was calculated as in Athalye et al.[7] using code adapted from Yu et al.[69]. Briefly, we used the factor analysis function *fastfa* to model each neuron's firing rate distribution as the sum of three elements: (1) a mean rate; (2) private variance from neuron-specific firing fluctuations not shared by the population being analyzed, and (3) shared variance driven by population signals (factors). For each neuron, shared over total variance is calculated as shared variance/(shared variance + private variance).

Using the *fastfa* function, we represent a column vector of $N$ neurons' mean firing rates as $\mu$ (rank $N$). Each neuron's private variance is an elements of the diagonal matrix $\Sigma^{\text{private}}$ (rank $N \times N$). Each neuron's weights for creating $k$ shared population signals are rows in the matrix U (rank $N \times k$, where $k < N$). The population covariance matrix is calculated as $\Sigma^{\text{shared}} = UU^{T}$ (rank $N \times N$), with the diagonal values representing the shared variance for each neuron. Therefore, the matrix of total variance for the population of neurons is represented as $\Sigma^{\text{total}} = \Sigma^{\text{shared}} + \Sigma^{\text{private}}$. Neuron $i$'s shared over total variance is $\Sigma^{\text{shared}}_{ii}/\Sigma^{\text{total}}_{ii}$. We chose to use $k = 3$ latent shared variables using the leave-one-out strategy outlined in Yu et al.[69].

**Subspace alignment**. The alignment between the subspaces defined by CCA and by PCA (Supplementary Fig. 4) was calculated using the MATLAB function *subspace*. Weights for the top three PCs were included for the local subspace. Weights for only the top one CV were included for the shared subspace.

**Statistical analysis**. Unless stated otherwise, all measurements were taken from distinct samples. We did not adjust for multiple comparisons. Normality was explicitly tested via Anderson–Darling test. For distributions that were non-normal, statistical testing was done using hierarchical bootstrap analysis, which is nonparametric[70] and $p$ values were computed as a one-sided test unless otherwise noted. For hierarchical bootstrap tests, statistics are written as mean ± standard deviation, and data was clustered by rat identity and, where relevant, condition (early vs. late, or baseline vs. infusion). Unless otherwise noted, $10^4$ permutations were used. For one-sided tests the lowest obtainable $p$ value was 0.0001, and for two-sided tests the lowest $p$ value obtainable was 0.0002; therefore, some $p$ values are reported as $p < 0.0001$ or $p < 0.0002$ rather than precise values. When distributions were normal, we used linear mixed-effect modeling by the MATLAB function *fitlme*. For mixed-effect models, statistics are written as mean ± standard error of the mean, and rat identity was always considered a random effect. When calculating changes in neural reach modulation between early and late learning, we included reach duration as a covariate to control for changes in reach duration between early and late learning. When calculating the relationship between neural reach modulation (log) and reach duration (log), only trials with positive neural modulation were included, and we included learning stage (early vs. late) as a covariate.

**Reporting summary**. Further information on research design is available in the Nature Research Reporting Summary linked to this article.

## Data availability
The datasets generated and analyzed in the current study are available from the corresponding author on reasonable request. Source data are provided with this paper.

## Code availability
The custom data analysis code created in MATLAB, and custom recording programs in OpenEx and Synapse (TDT) are available from the corresponding author on reasonable request.

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

## Acknowledgements

We thank Seok-Joon Won for assistance with perfusions and imaging. We also thank C. Campillo Rodriguez for assistance with histology; N. Hoglen, L. Tian, P. Khanna, and P. Shirvalkar for providing comments on the manuscript. This work was supported by fellowship awards from the National Defense Science and Engineering Graduate Fellowship (NDSEG, https://ndseg.asee.org/) the UCSF Discovery Fellows Program, and the Markowski-Leach Fellowship (to T.L.V.), the UCSF Medical Scientist Training Program (to T.L.V. and S.K.), the UCSF Neuroscience Graduate Program (to T.L.V. and K.D.), and the UCSF-UCB Bioengineering Graduate Program (to S.K.). Additional funds come from the Department of Veterans Affairs, Veterans Health Administration (VA Merit: 1I01RX001640 to K.G) and National Institute of Mental Health, NIH (5R01MH111871 to K.G.); and start-up funds from the UCSF Department of Neurology to K.G. K.G. also holds a Career Award for Medical Scientists from the Burroughs Wellcome Fund (1009855) and an Independent Scientist Award (1K02NS093014) from the National Institute of Neurological Disorders and Stroke, NIH. The funders had no role in study design, data collection and analysis, decision to publish, or preparation of the manuscript.

## Author contributions

T.L.V., K.D., and K.G. designed the study. T.L.V., K.D., and S.K. conducted experiments. T.L.V. and K.D. performed analyses. T.L.V., K.D., and K.G. wrote the manuscript.

## Competing interests

The authors declare no competing interests.
