## [Peer Review File · Nature Communications]

Reviewers' Comments:

Reviewer #1:

Remarks to the Author:

The paper by Veuthney, Dressier, and Ganguly is a nice investigation of long-term skill learning. The authors perform a series of interesting analyses to show that skill-learning in rats is paralleled by an increase in the coordination between premotor and motor cortical population activity. I also like their "causal" control showing that inactivation of premotor cortex disrupts this shared activity and leads to degraded task performance. The manuscript is well-written and the figures are clear, although I have some comments about the latter. I also have some questions/suggestions about some analyses and the introduction, as outlined below.

Comments:

1. I don't think the abstract stresses the novelties of the paper: We know (at least in primates) that premotor activity is predictive of primary motor activity (e.g., Kaufman et al Nature Neurosci 2014; Perich et al Neuron 2018); we also know that disrupting premotor activity with microstimulation — again, in primates— disrupts subsequent reaches (e.g., Churchland et al J Neurophysiol 2007). What I think is novel and interesting is that the authors' have studied long-term learning of how a skill is developed, and found an interesting neural correlate. I suggest they modify the abstract to highlight the central novel findings.

2. Lines 40-42, and elsewhere in the paper. Perhaps it is intentional, but I think the authors should differentiate between what they studied (long-term skill learning) vs. short-term learning or adaptation, which is what some of the references they cite report. Making such distinction is not only important to highlight what's novel in the paper, but also because short- and long-term learning may be quite different

3. Lines 99-101: "Over learning, significantly more neurons in both M2 and M1 were modulated by movement (quantifications in legend), consistent with the idea that learning engages and amplifies movement representations in both regions" -> Personally, I'm concerned about this type of statements. An increase in the number of modulated neurons after learning could be simply due to the animals performing the task in a more consistent manner: if a neuron had a fixed relationship with behavior, an increase in movement consistency would increase the "SNR" of the PETHs, and misleadingly point at a change in their relationship, even if the authors are aligning with respect to reach onset. The authors seem to agree with this, as per Lines 107-117. I thus suggest they rephrase these lines accordingly.

4. On the CCA and PCA methods:

i) Line 141, when the authors say "single high-dimensional population spaces," do they mean "area-specific high-dimensional population spaces"?

ii) A second difference between CCA and PCA is that the CCA axes do not need to be orthogonal among them, whereas those identified by PCA are —perhaps it is worth pointing this out;

iii) I couldn't find in the Methods what the R2 of the CVs is —apologies if I somehow missed it. Is it the error when reconstructing the activity of individual neurons from the CVs? Or simply the squared (canonical) correlation coefficient?

iv) Assuming that R2 is the reconstruction error, I don't know how to interpret the results in Fig S3a: Since using wider bins smooths out the fluctuations in the neural activity, is it better to have a mean R2 of 0.3 or 0.25? My guess is that using wider bins will increase the % variance explained by the top X CVs (and the CCs), but interpreting this type of results is always tricky. In my opinion, a more

interesting comparison is whether the directions identified with different bin sizes are similar. This could be achieved by computing pairwise angles between CCA axes or by comparing the subspaces they span using canonical (also called principal) angles (Gallego et al Nature Comm 2018)

v) The authors should include examples of the typical CCA plot showing the canonical correlation vs. the canonical variable (e.g., Fig 5 in Susillo et al Nature Neurosci 2015); as it is usually revealing of how many CVs may be significant.

vi) It seems to me that the authors' comparison between pairwise correlations and CCA weights provides further evidence that communication between cortical areas is better modeled based on population-wide activity patterns that are shared across the entire population rather than on interactions between single neurons. If the authors agree, I think this kind of phrasing would be useful.

vii) The authors use bootstrapping to assess how many CVs are significant. Regarding the procedure described in Lines 878-80: Did the authors separately shuffle the activity of each neuron across trials, or did they shuffle all the neurons at the same time? Did they do the shuffling for both areas or just one? My concern is whether their bootstrapping approach may be discarding a condition-independent signal (Kaufman et al, eNeuro 2016) as not significant.

viii) It would be nice if the authors dug a bit more into the shared M2-M1 latent signal: Is it a very good predictor of movement onset as the condition-independent signal was in the Kaufman paper (I suspect it could, based on Fig 8a)? Does it relate to reach kinematics at all, or are these better much predicted by local M1 activity?

ix) Computing angles between hyperplanes in high-dimensional spaces is tricky. The authors used the matlab function `subspace`, which computes one angle between the hyperplanes. However, to the best of my knowledge, the best way to compare two m-dimensional hyperplanes is to compute their m canonical or principal angles (Björk & Golub, Math Comput 1973). Do the conclusions of their Fig S4 if the authors adopt this approach? —Note that an alternative approach to compare two hyperplanes based on the variance they span is presented in (Elsayed, Lara et al Nature Comm 2016).

x) The authors computed the cross-area shared dynamics using CCA. An alternative approach, which would denoise their data, would be to find the dominant activity patterns in M1 and M2 using PCA, and then identify their crossed area interactions using CCA. Perhaps is something they want to try out?

5. I like the analysis in Fig 6, but I wonder what would happen if the authors computed CCs between the M2 and M1 signals with different lags (e.g., from -500 ms to +500 ms), would they get the same result that M2 activity consistently precedes M1 activity (based on higher correlations)?

6. Did the M1 manifold change significantly after M2 inactivation with muscimol? Did its dimensionality "shrink" (e.g., because of having very little variance in shared dimensions)? These points are relevant for their discussion on "on-manifold causal manipulation of downstream..."

- How does having the same mean firing rate and local variance imply that you're on the same manifold? Or do the authors mean the same "local manifold", i.e. the dimensions that are private and not shared?

Other remarks:

-
- Lines 20-21: "Prior work has focused on local population dynamics"; the authors cite at least one paper that studies cross-area population dynamics during movement adaptation (Perich et al Neuron 2018); perhaps they mean that "Prior studies of long-term learning / skill acquisition have focused ..."?
 - Line 23: "motor (M1) and premotor (M2) cortices" -> "rodent motor (M1) ..." To avoid misunderstandings with primate PMd/PMv.
 - Line 39 and elsewhere: in monkeys, premotor cortex is denoted as PM, and spans dorsal (PMd) and ventral (PMv) sites). I suggest the authors adopt this nomenclature throughout the paper to avoid

misunderstandings between models

- Line 51: "patterns of dominant variance" -> PCA finds dominant co-variance patterns
- Line 92: "3.61s Hz \pm 0.36 " -> delete "Hz"
- Line 127: I suggest the authors to avoid headings based on a specific method, and rather stress the main finding —after all, other methods could identify relationships similar to those revealed by CCA.
- Fig S3: "CCA activity" doesn't seem very accurate mathematically; perhaps use "canonical variable amplitude" or something like shared activity...
- Fig 2e: "Neurons are ordered by the absolute value of their CCA weight" -> meaning the weights onto the first CV? or the norm 1 of the weights onto all significant CVs? I'm having problems to see the said correlation between the mean CCA weights and the short-latency normalized cross-correlation values; please add a scatter plot to see the correlation
- Fig 2f: Add correlation value between the M2 and M1 canonical variables, to have a sense of how similar they are
- Fig 3 (line 688): " across behavioral states" -> "across behaviors"
- Fig 5: I was a bit confused about the colors. In (a) and (b) you use grey for baseline and gold for movement, but then in (c) gold means late cross-area modulation; I'd use different colors in (c) than black and gold. Also, in (d) I'd color the "early" and "late" bars as in (c) —now they are all colored as in "late". What are the grey lines in (d), individual animals?
- Fig 8: I found the colors a bit confusing for similar reasons —gold and grey in (a) indicate different things than in (b). Also, what are the yellow lines in (d), individual mice? Same for the grey lines in (f)
- Lines 340-1: "This study outlines a new approach to understanding activity shared between two nodes in a neural network." -> sounds a bit vague, why not "two cortical areas"?
- Line 345: "M1 cross-area population dynamics" -> "M1-M2" ?
- Lines 364-5: "but they are rarely combined to understand how cross area versus local signals contribute to motor network function." -> as the authors point out in the Introduction, this is done in their Ref. 5
- "Axes of variance" -> "patterns of co-variance" That's what the authors are capturing.
- Lines 457-8: I think that this first sentence could be stronger
- Lines 939-40: clarify
- "Neural data analysis: mean local covariance": this summary isn't very clear. What are N and z in the dimensions of U? How is the matrix R computed? Please expand
- I'm not very familiar with the MATLAB's function 'fitlme', but I think the authors should either check for normality or use a non-parametric method. What am I missing?

Reviewer #2:

The authors apply a series of sophisticated analyses on extracellularly recorded neural responses to unravel the cross-area population dynamics during the learning and execution of skilled reaching movements. The analyses employed are sound, and the results are well-presented both in writing and in illustrations. The modular organization of the manuscript helps the reader to follow the rationale of the analyses.

My main objection concerns the cerebral cortical areas from which the analyzed activity was recorded. The authors claim that the activity has been recorded from the motor (M1) and premotor (M2) cortex. However, they do not provide any evidence to support the selection of the coordinates for the electrode implantation. Given the inter-subject variability, the recorded hemisphere of each animal should had been mapped electrophysiologically (intracortical microstimulation) to reveal the extent of the forelimb representations in each of the two cortical areas and guide the implantation of the electrode arrays in each subject. The electrode localization provided in Fig. S1 does not support authors' claim that the activity has been recorded from the motor (M1) and premotor (M2) cortex.

Inspection of motor representation maps of the rat cortex as well as brain atlases reveals that the electrodes aiming M1 were located too lateral, at best at the M1/S1 transition and those aiming M2 were too anterior. Therefore, there is a great risk that the activity analyzed has been recorded either from adjacent cortical area or adjacent motor representations. Under these circumstances the M1/M2 cross-area dynamics argument is very weak.

OTHER ISSUES

Fig. 1: The text for c & d in the legend is in disagreement with the text on the figure. Top/bottom on the figure refers to M2/M1 not to early/late learning (which is left/right on the figure).

PHARMACOLOGICAL INFUSIONS: (a) Did the different infused volumes (0.5-1uL) evoked different effects? Please provide data in support of either case. (b) Please provide histological verification of the spread of the injections. (c) Is the infusion rate reported correctly? (d) The existence of a control injection of muscimol at a nearby location to verify the specificity of the effect obtained following the injection in the presumed M2 location would be a very elegant additional control.

HISTOLOGY: Please provide histological verification (stained sections demonstrating gliosis/electrolytic lesions) for the implanted electrodes whatever their location is. The section drawings with the ovals are not convincing.

BEHAVIORAL TRAINING: Please provide more details about the training. How many trials/time were the animals allowed to complete the "at least 30 trials" required to define the "early learning" training day? How many days were necessary to reach the "late learning" day?

BEHAVIORAL ANALYSIS: The accuracy of the kinematic data (maximum movement speed, reach onset, grasp onset) obtained with the 30 Hz camera is questionable. This could severely affect the analyses performed and consequently the results obtained.

How many animals were video recorded with the 30 Hz camera and how many with the 100 Hz one? How similar are the data obtained with the two types of camera? Please provide a table with all the kinematic data used for the analyses (for each of the subjects). The kinematic data provided at Fig. S2 are not sufficient.

Reviewer #3:

Remarks to the Author:

The current study by Veuthey et al. represents a fascinating addition to the literature regarding the relationship between M2 and M1 in the rat during motor skill learning. They were able to show how activity within M2 and M1 changes after skill learning, and importantly how the cross-area coupling between M2 and M1 changes with learning. The most powerful finding is that when comparing baseline and inactivation of M2 within the same day, M2 inactivation selectively affects the M2-M1 cross-area dimensions, while seemingly leaving the local structure within M1 largely intact.

Congratulations on a great manuscript! Overall, I find the manuscript to be well-written and convincing. I propose a few changes for the purpose of clarification about methodological detail and overall logic.

1) Most results rely very heavily on the CCA approach. One thing I felt was lacking was an explanation of why this method was better than similar methods used in previous work. For example, why not the GLM approach used in Perich et al. (2018)? Why not the granger causality method in Makino et al. (2017)?

2) As someone who has worked with CCA in the past, I have generally found it inappropriate for identifying meaningful ("causal") dimensions in neural population data, and a number of studies

recommend alternative methods (see Kornblith et al., 2019) that are more reliable. Maybe the fact that you performed CCA on single trials across areas was able to compensate for this? Can you confirm that CCA was carried out on simultaneous recordings from M2 and M1, since it wasn't completely clear. A bit more explanation would be useful here, since I was not completely convinced by the cross-validation procedure showing less variability in weights across folds than overall.

3) Have you considered that finding the optimal cross-area dimensions may require a time lag between areas? Presumably this could be answered by inspecting the pair-wise cross-correlograms you calculated.

Very minor comment:

- In some cases there are spaces between words and the corresponding reference number, no doubt a remnant of submission to a journal that used a different reference style.

References

Kornblith, S., Norouzi, M., Lee, H., and Hinton, G. (2019). Similarity of Neural Network Representations Revisited.

Makino, H., Ren, C., Liu, H., Kim, A.N., Kondapaneni, N., Liu, X., Kuzum, D., and Komiyama, T. (2017). Transformation of Cortex-wide Emergent Properties during Motor Learning. *Neuron* 94, 880-890.e8.

Perich, M.G., Gallego, J.A., and Miller, L.E. (2018). A Neural Population Mechanism for Rapid Learning. *Neuron* 100, 964-976.e7.

General Response	1
Specific Response to Reviewer #1 Comments	1
Specific Response to Reviewer #2 Comments	10
Specific Response to Reviewer #3 Comments	14

General Response

We sincerely thank the reviewers for their in-depth review of our manuscript. We appreciate their summary assessments that we performed “a nice investigation of long-term skill learning”, that “the analyses employed are sound, and the results are well-presented both in writing and in illustrations”, and that our manuscript “represents a fascinating addition to the literature regarding the relationship between M2 and M1 in the rat during motor skill learning.” The reviewer’s thoughtful comments and suggestions have guided us to extensively revise the previous draft of the manuscript, carry out several new analyses, and perform an additional experiment that we believe have significantly strengthened our study and its presentation through the manuscript.

Specific Response to Reviewer #1 Comments

The paper by Veuthey, Derosier, and Ganguly is a nice investigation of long-term skill learning. The authors perform a series of interesting analyses to show that skill-learning in rats is paralleled by an increase in the coordination between premotor and motor cortical population activity. I also like their “causal” control showing that inactivation of premotor cortex disrupts this shared activity and leads to degraded task performance. The manuscript is well-written and the figures are clear, although I have some comments about the latter. I also have some questions/suggestions about some analyses and the introduction, as outlined below.

We thank Reviewer #1 for their comments.

Comments:

1. I don’t think the abstract stresses the novelties of the paper: We know (at least in primates) that premotor activity is predictive of primary motor activity (e.g., Kaufman et al Nature Neurosci 2014; Perich et al Neuron 2018); we also know that disrupting premotor activity with microstimulation —again, in primates— disrupts subsequent reaches (e.g., Churchland et al J Neurophysiol 2007). What I think is novel and interesting is that the authors’ have studied long-term learning of how a skill is developed, and found an interesting neural correlate. I suggest they modify the abstract to highlight the central novel findings.

We thank the reviewer for their suggestion to modify the abstract to highlight the central novel findings more clearly. We have edited the abstract with this in mind.

2. Lines 40-42, and elsewhere in the paper. Perhaps it is intentional, but I think the authors should differentiate between what they studied (long-term skill learning) vs. short-term learning or adaptation, which is what some of the references they cite report. Making such distinction is not only important to highlight what's novel in the paper, but also because short- and long-term learning may be quite different.

We thank the reviewer for emphasizing the difference between long term skill learning and short-term learning or adaptation. Per their suggestion, we have modified the manuscript title, the abstract line 23, and lines 36-38 and 39-41.

3. Lines 99-101: “Over learning, significantly more neurons in both M2 and M1 were modulated by movement (quantifications in legend), consistent with the idea that learning engages and amplifies movement representations in both regions” -> Personally, I'm concerned about this type of statements. An increase in the number of modulated neurons after learning could be simply due to the animals performing the task in a more consistent manner: if a neuron had a fixed relationship with behavior, an increase in movement consistency would increase the “SNR” of the PETHs, and misleadingly point at a change in their relationship, even if the authors are aligning with respect to reach onset. The authors seem to agree with this, as per Lines 107-117. I thus suggest they rephrase these lines accordingly.

We thank the reviewer for raising this important point. We have revised lines 100-113 accordingly.

4. On the CCA and PCA methods:

i) Line 141, when the authors say “single high-dimensional population spaces,” do they mean “area-specific high-dimensional population spaces”?

We thank the reviewer for this comment. We have changed the language in that sentence accordingly (lines 135).

ii) A second difference between CCA and PCA is that the CCA axes do not need to be orthogonal among them, whereas those identified by PCA are —perhaps it is worth pointing this out.

Thank you for bringing this up. As the reviewer points out, PCA axes are orthogonal, while canonical variables produced by CCA are constrained to be uncorrelated (but not necessarily orthogonal). However, since a large portion of our analyses focus on the top CV only, and we bring up PCA only as a brief reference to a method many neuroscientists will be more familiar with, we respectfully suggest that a complete comparison of the differences between the two methods is beyond the scope of this work.

iii) I couldn't find in the Methods what the R² of the CVs is —apologies if I somehow missed it. Is it the error when reconstructing the activity of individual neurons from the CVs? Or simply the squared (canonical) correlation coefficient?

We thank the reviewer for highlighting this lack of clarity. The R^2 we use comes from cross-validation, and is intended to measure how well the models generalize to held-out data. Specifically, we randomly partition the full dataset into 10 folds (ignoring trial structure, i.e. adjacent timepoints in the same trial may be assigned to different folds). Then (ten times) we assign one fold to be the test data and the other nine to be the training data. We fit a CCA model to the training data, then project the test data onto this model, and compute the R^2 between the M1 and M2 projections. We have expanded and clarified the relevant section in the Methods, lines 552-563.

iv) Assuming that R^2 is the reconstruction error, I don't know how to interpret the results in Fig S3a: Since using wider bins smooths out the fluctuations in the neural activity, is it better to have a mean R^2 of 0.3 or 0.25? My guess is that using wider bins will increase the % variance explained by the top X CVs (and the CCs), but interpreting this type of results is always tricky. In my opinion, a more interesting comparison is whether the directions identified with different bin sizes are similar. This could be achieved by computing pairwise angles between CCA axes or by comparing the subspaces they span using canonical (also called principal) angles (Gallego et al Nature Comm 2018).

We thank the reviewer for this suggestion. As noted above, in our analysis, the R^2 is not the error when reconstructing the activity of individual neurons, but rather a measure of generalizability to held out data. However, we agree that the question of whether the models are identifying similar directions of covariance is interesting. We computed the angle between the top CV for models fit to 100ms, 75ms, and 50ms binned data, and found that for both M1 and M2, the median angle between 100ms models and 75ms models was smaller than the angle between 100ms models and 50ms models, or between 75ms models and 50ms models. Hierarchical bootstrapping indicated that this difference was significant except in one case: in M1, the angle between the 100ms and 75ms models was not significantly smaller than the angle between 75ms models and 50ms models. In all cases, the angle between models was relatively small (10-20 deg). Our interpretation is that all CCA models fitted at this range of binwidths are identifying similar patterns of covariance, but that models with closer binwidths are more similar. Please see Figure S3 and Methods lines 580-601.

v) The authors should include examples of the typical CCA plot showing the canonical correlation vs. the canonical variable (e.g., Fig 5 in Susillo et al Nature Neurosci 2015); as it is usually revealing of how many CVs may be significant.

We thank the reviewer for this suggestion. We have added the suggested plot to Figure S3. For both the canonical correlation and our cross-validated R^2 (described above in response to point 4.iii), values drop off more sharply than in Susillo et al., and seem to be fairly well in agreement with the number of CVs determined as significant via our trial-shuffle method (described below in response to point 4.vii).

vi) It seems to me that the authors' comparison between pairwise correlations and CCA weights provides further evidence that communication between cortical areas is better modeled based on population-wide activity patterns that are shared across the entire

population rather than on interactions between single neurons. If the authors agree, I think this kind of phrasing would be useful.

We thank the reviewer for highlighting this point. We have incorporated this kind of phrasing in lines 170-172.

vii) The authors use bootstrapping to assess how many CVs are significant. Regarding the procedure described in Lines 878-80: Did the authors separately shuffle the activity of each neuron across trials, or did they shuffle all the neurons at the same time? Did they do the shuffling for both areas or just one? My concern is whether their bootstrapping approach may be discarding a condition-independent signal (Kaufman et al, eNeuro 2016) as not significant.

We thank the reviewer for these comments and clarified our methods accordingly (lines 567-576), in particular emphasizing that we did shuffle all neurons in each region together. We agree that our approach may discard a condition-independent signal; in fact, this was an intentional design choice. While we believe this condition-independent signal to be interesting and important, one of our goals was to test whether our CCA model could capture signals specific to single-trial behavior, which it did robustly. M1 and M2 are known to have similar subcortical outputs, and our results and others demonstrate that many neurons in both regions show large fluctuations in firing rate at the time of grasp onset. With this in mind, we reasoned that canonical variables that described moment-by-moment correlations above and beyond the correlations expected based on the large fluctuations at grasp onset would be more likely to more strictly capture underlying causal interactions between the regions. For these reasons, while we agree that a condition-independent signal would be interesting in its own right, for our purposes we did not attempt to preserve it.

viii) It would be nice if the authors dug a bit more into the shared M2-M1 latent signal: Is it a very good predictor of movement onset as the condition-independent signal was in the Kaufman paper (I suspect it could, based on Fig 8a)? Does it relate to reach kinematics at all, or are these better much predicted by local M1 activity?

We thank the reviewer for this comment. Indeed, the M1-M2 latent signal is a good predictor of movement onset, and this prediction is improved with learning, as illustrated in Figure 4. Additionally, the M1-M2 latent signal predicts reach duration, and this prediction is improved with learning, as illustrated in Figure 5.

ix) Computing angles between hyperplanes in high-dimensional spaces is tricky. The authors used the matlab function `subspace`, which computes one angle between the hyperplanes. However, to the best of my knowledge, the best way to compare two m-dimensional hyperplanes is to compute their m canonical or principal angles (Björk & Golub, Math Comput 1973). Do the conclusions of their Fig S4 if the authors adopt this approach? —Note that an alternative approach to compare two hyperplanes based on the variance they span is presented in (Elsayed, Lara et al Nature Comm 2016).

We thank the reviewer for these suggestions. To our knowledge, the Matlab `subspace` function is an implementation of the algorithm from (Björk & Golub, Math Comput 1973) that returns the

first principal angle, which we find to be the most informative. We reviewed the variance metric from (Elsayed, Lara et al Nature Comm 2016), which seems excellent for those authors' purpose, which was to compare two sets of weights from different time periods fitted to optimize local variance using PCA. However, our purpose was instead to compare two sets of weights from the same time period fitted to optimize either local variance (using PCA) versus shared variance (using CCA). To us, these key differences in the goals of the analyses means that the method used in Elsayed et al would need to be heavily modified to be applicable to our data, and would suffer from additional caveats in interpretation. We thank the reviewer for this suggestion.

x) The authors computed the cross-area shared dynamics using CCA. An alternative approach, which would denoise their data, would be to find the dominant activity patterns in M1 and M2 using PCA, and then identify their crossed area interactions using CCA. Perhaps is something they want to try out?

We thank the review for this comment. This suggested approach would de-noise the data and allow for comparison of dominant activity patterns in M1 and M2. However, one of our goals was to capture moment-by-moment shared activity between M1 and M2 which might differ across trials and therefore might have been eliminated through single-area PCA-based de-noising. This concern was especially important in light of our early learning data, where the behavior is quite variable on different trials. Specifically, comparing PCA-based models of M1 and M2 may make it more difficult to identify how single-trial M1-M2 shared activity relates to behavior. In our opinion, the suggested approach would be more appropriate for a dataset with more stereotyped behavior.

5. I like the analysis in Fig 6, but I wonder what would happen if the authors computed CCs between the M2 and M1 signals with different lags (e.g., from -500 ms to +500 ms), would they get the same result that M2 activity consistently precedes M1 activity (based on higher correlations)?

We thank the reviewer for this suggestion. In terms of single-unit cross-correlations, we found that many neuron pairs had highest cross-correlation when the M2 neuron lead, but across the population as a whole, hierarchical bootstrapping found that the optimal timelag for pairwise cross-correlation was not significantly different from zero. As an additional way to investigate timing, we fit CCA models for data binned at 50, 75, and 100ms, and with lags from -500ms to +500ms. For 6 of 8 datasets, using 100ms time bins with no lag provided the highest correlation between M1 and M2 neural activity (see Methods lines 582-603). We think these analyses support our conclusion that the most correlated *population signals* shared between the regions do not have a large timelag between M2 and M1. However, we wish to acknowledge that our population signals are computed off of very sparse samplings of neurons from each region. Given that single-synapse connections do exist between M2 and M1, it is likely that a more complete sampling of M2 and M1 signals would reveal shared activity between M2 and M1 neurons with small time lags. Additionally, in Figure 6, we highlight that local M1 and M2 population signals are less correlated between the regions and have a significant timelag in which M2 precedes M1.

6. Did the M1 manifold change significantly after M2 inactivation with muscimol? Did its dimensionality “shrink” (e.g., because of having very little variance in shared dimensions)? These points are relevant for their discussion on “on-manifold causal manipulation of downstream...” How does having the same mean firing rate and local variance imply that you’re on the same manifold? Or do the authors mean the same “local manifold”, i.e. the dimensions that are private and not shared?

We thank the reviewer for these questions. Unfortunately, the number of M1 neurons available from each animal does not permit in-depth analysis of manifold dimensionality before and after M2 inactivation. As pointed out, having the same mean firing rate and local variance *cannot* prove that the data lies on the same manifold. (Also of note here is that when tested via hierarchical bootstrap instead of mixed effect model, the small decrease in M1 firing rate with M2 muscimol was significant, see our response to the remark about normality testing below.) To reflect that we are not discussing manifolds in the strict mathematical definition of the term, we have eliminated use of the word ‘manifold’ to instead highlight that gross local shared variance in M1 is not changed by M2 inactivation (lines 405-407, 412-413). We believe this supports the idea that the behavioral changes in skilled reaching after M2 inactivation are not due to gross disruption of M1, but rather a decoupling from M2, leading to a decrease in learned movement coordination.

Other remarks:

- Lines 20-21: “Prior work has focused on local population dynamics”; the authors cite at least one paper that studies cross-area population dynamics during movement adaptation (Perich et al Neuron 2018); perhaps they mean that “Prior studies of long-term learning / skill acquisition have focused ...”?

We thank the reviewer for this suggestion, which has been implemented.

- Line 23: “motor (M1) and premotor (M2) cortices” -> “rodent motor (M1) ...” To avoid misunderstandings with primate PMd/PMv.

We thank the reviewer for this suggestion, which has been implemented.

- Line 39 and elsewhere: in monkeys, premotor cortex is denoted as PM, and spans dorsal (PMd) and ventral (PMv) sites). I suggest the authors adopt this nomenclature throughout the paper to avoid misunderstandings between models.

We thank the reviewer for this suggestion. We recognise that there is important literature distinguishing PMv and PMd in non-human primates. In this manuscript, we reference and attempt to integrate findings, interpretations, and models from studies conducted with both non-human primates and rodents. Since PMv and PMd are not well distinguished in rodents, we found it more approachable to use the more generic terms ‘PM’ and M2. Thank you.

- Line 51: “patterns of dominant variance” -> PCA finds dominant co-variance patterns.

We thank the reviewer for catching this error, which has been corrected.

- Line 92: “3.61s Hz \pm 0.36 “ -> delete “Hz”

We thank the reviewer for catching this error, which has been corrected.

- Line 127: I suggest the authors to avoid headings based on a specific method, and rather stress the main finding —after all, other methods could identify relationships similar to those revealed by CCA.

We thank the reviewer for this suggestion, which has been implemented. (line 122)

- Fig S3: “CCA activity” doesn’t seem very accurate mathematically; perhaps use “canonical variable amplitude” or something like shared activity...

We thank the reviewer for this suggestion, which has been implemented.

- Fig 2e: “Neurons are ordered by the absolute value of their CCA weight” -> meaning the weights onto the first CV? or the norm 1 of the weights onto all significant CVs? I’m having problems to see the said correlation between the mean CCA weights and and the short-latency normalized cross-correlation values; please add a scatter plot to see the correlation

We thank the reviewer for this comment. Yes, “the absolute value of their CCA weight” indeed refers to the weight onto the first CV, we have clarified this in the caption. Additionally, we have added a panel showing the scatter plot of short-latency normalized cross-correlation and absolute weight onto the first CV across all animals.

- Fig 2f: Add correlation value between the M2 and M1 canonical variables, to have a sense of how similar they are.

We thank the reviewer for this comment. Figure 2f is now figure 2g; we have added the R^2 value for the example trial to the caption (it is 0.3733). The R^2 values between M2 and M1 during spontaneous behavior, early learning reaches, and late learning reaches are also available in Figure 3b.

- Fig 3 (line 688): “ across behavioral states” -> “across behaviors”

We thank the reviewer for this suggestion, which has been implemented.

- Fig 5: I was a bit confused about the colors. In (a) and (b) you use grey for baseline and gold for movement, but then in (c) gold means late cross-area modulation; I’d use different colors in (c) than black and gold. Also, in (d) I’d color the “early” and “late” bars as in (c) —now they are all colored as in “late”. What are the grey lines in (d), individual animals?

We thank the reviewer for this comment. Figure 5c has been changed so that early learning is in black and late learning is in grey. The colors in Figure 5d were left in yellow as they correspond to values based on the yellow areas in Figure 5a. The grey lines in Figure 5d are indeed the individual animals. This information has been added to the figure legend.

- Fig 8: I found the colors a bit confusing for similar reasons —gold and grey in (a) indicate different things than in (b). Also, what are the yellow lines in (d), individual mice? Same for the grey lines in (f)

We thank the reviewer for these comments. The colors in Figure 8b and 8c have been changed so that data from pre-reach is black and data from reach start is grey. For Figures 8d and 8f, the legend has been modified to clarify which lines represent data from individual animals.

- Lines 340-1: “This study outlines a new approach to understanding activity shared between two nodes in a neural network.” -> sounds a bit vague, why not “two cortical areas”?

We thank the reviewer for this suggestion, which has been implemented (lines 336-337).

- Line 345: “M1 cross-area population dynamics” -> “M1-M2” ?

We thank the reviewer for this suggestion. These lines have been clarified (lines 293).

- Lines 364-5: “but they are rarely combined to understand how cross area versus local signals contribute to motor network function.” -> as the authors point out in the Introduction, this is done in their Ref. 5.

We thank the reviewer for highlighting this oversight. The reference has been added (line 356).

- “Axes of variance” -> “patterns of co-variance” That’s what the authors are capturing.

We thank the reviewer for this suggestion, which has been implemented (line 411).

- Lines 457-8: I think that this first sentence could be stronger

We thank the reviewer for this comment. The sentence in question has been edited (lines 422-423).

- Lines 939-40: clarify

We thank the reviewer for this comment, which highlighted a line in the methods which should have been deleted. This section has been clarified (line 646).

- “Neural data analysis: mean local covariance”: this summary isn’t very clear. What are N and z in the dimensions of U? How is the matrix R computed? Please expand.

We thank the reviewer for this comment. This section has been expanded. (lines 653-668)

- I’m not very familiar with the MATLAB’s function ‘fitlme’, but I think the authors should either check for normality or use a non-parametric method. What am I missing?

We thank the reviewer for this comment. We checked for normality in all analyses which used fitlme. Where necessary, we replaced those analyses with a hierarchical method based on bootstrap sampling, as outlined in Reference 73 (Saravanan et al., 2019). One result was impacted by this change: with mixed effect modeling, we had found that across animals the decrease in M1 firing rate with M2 muscimol was not significant, but with hierarchical bootstrapping we found that this decrease was significant. However, the addition of histological

verification of muscimol spread (Fig. S6) more directly supports our original point, which is that the change in M1 firing rate is not due to a direct effect of muscimol on M1 neurons.

Specific Response to Reviewer #2 Comments

The authors apply a series of sophisticated analyses on extracellularly recorded neural responses to unravel the cross-area population dynamics during the learning and execution of skilled reaching movements. The analyses employed are sound, and the results are well-presented both in writing and in illustrations. The modular organization of the manuscript helps the reader to follow the rationale of the analyses.

We thank the reviewer for their thoughtful comments and review of our manuscript.

My main objection concerns the cerebral cortical areas from which the analyzed activity was recorded. The authors claim that the activity has been recorded from the motor (M1) and premotor (M2) cortex. However, they do not provide any evidence to support the selection of the coordinates for the electrode implantation. Given the inter-subject variability, the recorded hemisphere of each animal should have been mapped electrophysiologically (intracortical microstimulation) to reveal the extent of the forelimb representations in each of the two cortical areas and guide the implantation of the electrode arrays in each subject. The electrode localization provided in Fig. S1 does not support authors' claim that the activity has been recorded from the motor (M1) and premotor (M2) cortex. Inspection of motor representation maps of the rat cortex as well as brain atlases reveals that the electrodes aiming M1 were located too lateral, at best at the M1/S1 transition and those aiming M2 were too anterior. Therefore, there is a great risk that the activity analyzed has been recorded either from adjacent cortical area or adjacent motor representations. Under these circumstances the M1/M2 cross-area dynamics argument is very weak.

We thank the reviewer for these comments. We agree that mapping the recorded hemisphere of each animal with intracortical microstimulation would have provided excellent information for electrode array placement to specifically target forelimb regions. Unfortunately, it is impossible for us to do this retrospectively on the experimental animals. Consequently, we have edited our manuscript and figures to include more references to prior papers which showed forelimb signals at similar coordinates for M1 and M2 for Long-Evans rats, specifically. Also, we performed intracranial microstimulation experiments on 2 additional animals, one with an array implanted in M1 and one with an array implanted in M2, and found that stimulation on multiple channels evoked forelimb movement in both animals. Further information on these stimulation experiments can be found in Methods, lines 453-469, and our implantation coordinates are given with more detail in Supplemental Table 2 and compared to our stimulation animals and past studies in Supplemental Figure 1. Given the consistency in these results, and the robust reaching-evoked neural responses we observed, we find it reasonable to assume that the experimental animals also had motor forelimb representations at the provided coordinates.

OTHER ISSUES

Fig. 1: The text for c & d in the legend is in disagreement with the text on the figure. Top/bottom on the figure refers to M2/M1 not to early/late learning (which is left/right on the figure).

We thank the reviewer for identifying this. The error has been corrected.

PHARMACOLOGICAL INFUSIONS: (a) Did the different infused volumes (0.5-1uL) evoked different effects? Please provide data in support of either case. (b) Please provide histological verification of the spread of the injections. (c) Is the infusion rate reported correctly? (d) The existence of a control injection of muscimol at a nearby location to verify the specificity of the effect obtained following the injection in the presumed M2 location would be a very elegant additional control.

We thank the reviewer for these comments.

- (a) The infused volumes were titrated for each animal. We first started with the larger volume (1uL). If the animal was unable to reach within 2 hours, we downscaled to the smaller volume (0.5 uL). This has been clarified in the Methods lines 471-475.
- (b) Unfortunately, removing extensive, closely-packed cortical hardware (2 probes and a cannula) from experimental animals damaged the brain tissue and prevented high-quality imaging to calculate muscimol spread (see image below). Consequently, spread was calculated from two additional animals in whom we infused muscimol through a Hamilton needle two hours before perfusion. Images of spread from one of these example animals is provided in Supplemental Figure 6. Additionally, a schematized image of muscimol spread from all available animals is compared with implantation sites in Supplemental Figure 1.

M2 slice from one of the muscimol + neural recording animals (T336). Fluorescent muscimol is shown in green, DAPI in blue. Post-mortem damage from removing probes and cannula prevents accurate assessment of muscimol spread.

- (c) There was a typo in the reported infusion rate, which has been corrected. Thank you.
- (d) We thank the reviewer for this thoughtful suggestion about infusion at unrelated nearby sites. While we agree that this would be an elegant experiment, it seems to be more about mapping the anatomical boundaries of M2. We respectfully suggest that this is out of the scope of our study. Our main goal here is to conduct simultaneous monitoring of ensemble dynamics across M2 and M1. We hope that our additional experiments have shown that our recordings are indeed in M2.

HISTOLOGY: Please provide histological verification (stained sections demonstrating gliosis/electrolytic lesions) for the implanted electrodes whatever their location is. The section drawings with the ovals are not convincing.

We thank the reviewer for these comments. We performed histological imaging from most animals (one died prior to perfusion). Although we observed gliosis and/or electrolytic lesions in some cases, in many cases, the main indication of electrode location was post-mortem tissue damage due to hardware removal, especially in animals with both cannulas and arrays. For example, the image below shows an M1 slice and an M2 slice from the same example animal (T131). Electrode tracks are visible in the M1 slice, but the M2 slice shows extensive post-mortem damage. In all cases, the indications of electrode placement we found were consistent with the surgical coordinates, which are now reported in greater detail in Supplemental Figure 1 and Supplemental Table 2.

M1 (left) and M2 (right) slices from the same animal (T131).

BEHAVIORAL TRAINING: Please provide more details about the training. How many trials/time were the animals allowed to complete the “at least 30 trials” required to define the “early learning” training day? How many days were necessary to reach the “late learning” day?

We thank the reviewer for these comments. We have included animal-specific data for these and other behavioral metrics in Supplemental Table 1.

BEHAVIORAL ANALYSIS: The accuracy of the kinematic data (maximum movement speed, reach onset, grasp onset) obtained with the 30 Hz camera is questionable. This could severely affect the analyses performed and consequently the results obtained. How many animals were video recorded with the 30 Hz camera and how many with the 100 Hz one? How similar are the data obtained with the two types of camera? Please provide a table with all the kinematic data used for the analyses (for each of the subjects). The kinematic data provided at Fig. S2 are not sufficient.

We thank the reviewer for these comments. We agree that kinematic data obtained at a higher framerate is generally preferable. By focusing our kinematic analyses on time of reach onset (Fig. 4) and reach duration (Fig. 5), we have restricted ourselves to metrics where the 30 Hz data appeared sufficient. Moreover, our results obtained using higher framerates did not appear to affect our conclusion. We have included animal-specific kinematic data in Supplemental Table 1.

Specific Response to Reviewer #3 Comments

The current study by Veuthey et al. represents a fascinating addition to the literature regarding the relationship between M2 and M1 in the rat during motor skill learning. They were able to show how activity within M2 and M1 changes after skill learning, and importantly how the cross-area coupling between M2 and M1 changes with learning. The most powerful finding is that when comparing baseline and inactivation of M2 within the same day, M2 inactivation selectively affects the M2-M1 cross-area dimensions, while seemingly leaving the local structure within M1 largely intact.

Congratulations on a great manuscript! Overall, I find the manuscript to be well-written and convincing. I propose a few changes for the purpose of clarification about methodological detail and overall logic.

We thank the reviewer for their supportive and enthusiastic comments.

1) Most results rely very heavily on the CCA approach. One thing I felt was lacking was an explanation of why this method was better than similar methods used in previous work. For example, why not the GLM approach used in Perich et al. (2018) ? Why not the granger causality method in Makino et al. (2017)?

We thank the reviewer for these comments. We agree that each method has its strengths and weaknesses, and that many other methods would have been interesting to apply to this dataset. In lines 69-74, we added clarification and emphasis on the reasons for which we chose to use CCA. Specifically, we chose CCA because it is designed to look for correlated population readouts and it reduces dimensionality and optimizes for M2-M1 covariation simultaneously rather than sequentially. Importantly, this allows for the possibility of identifying cross-area signals that may be missed by other methods which first reduce dimensionality and then look for covariations, as highlighted in lines 54-58.

In Perich et al. (2018), the authors combine PCA and GLMs to build models which predict M1 spiking from PMd PCs. However, by using PCA on the PMd data before relating it to M1, this method assumes that signals communicated between PMd and M1 dominate PMd local variance and can be identified by looking at local patterns of variance in PMd alone. This assumption is more appropriate for the behavioral paradigm used in Perich et al. because their animals were extensively trained and produced highly stereotyped behavior prior to the perturbations. We believe that this extensive training sets up robust cross-area signals specifically related to the trained behaviors, making it more likely that local signals extracted with PCA also reflect communicated cross-area signals. In contrast, our dataset reflects motor signals that evolve from early to late learning of new skilled movements, at timepoints during which cross-area signals may not yet dominate local variance sufficiently well to be identified by dimensionality reduction method such as PCA that optimizes local properties. Indeed, when comparing axes of cross-area covariance identified by CCA with axes of local variance identified

by PCA (Figure S4), we found that the two methods resulted in different patterns of weights assigned to the units, and identified subspaces with a large angle between them, suggesting that, in our dataset, CCA finds patterns of cross-area covariance that might be obscured by approaches that starts with PCA.

In Makino et al. 2017, they analyze wide-field calcium imaging signals. Applying Granger Causality to these continuous bulk signals allows them to estimate a timelag and a direction of causality between M2 and M1. However, this method is limited to identifying only cross-area signals which are able to dominate the local calcium signal. Applying CCA to spiking data allows us to identify and analyze cross-area signals in more detail.

2) As someone who has worked with CCA in the past, I have generally found it inappropriate for identifying meaningful (“causal”) dimensions in neural population data, and a number of studies recommend alternative methods (see Kornblith et al., 2019) that are more reliable. Maybe the fact that you performed CCA on single trials across areas was able to compensate for this? Can you confirm that CCA was carried out on simultaneous recordings from M2 and M1, since it wasn’t completely clear. A bit more explanation would be useful here, since I was not completely convinced by the cross-validation procedure showing less variability in weights across folds than overall.

We thank the reviewer for these expert comments. The methods in Kornblith et al. (2019) are very interesting, and we would be eager to apply them to future work. Additionally, we agree with their main criticism of CCA, namely the need to limit analyses to very few dimensions relative to the amount of data. Here we limited our analysis to significant CCA dimensions (only 1-3 dimensions depending on the animal), as identified through cross-validation and trial-shuffling (see expanded Methods lines 567-576). We would also like to confirm that CCA was carried out on simultaneous recordings of M2 and M1, and we further emphasised this in line 23 (Abstract) and lines 70, 72, 88, 124,281, 336, 358. We have also clarified our remarks on weight variability across folds (lines 141-147). Specifically, we think this analysis helps confirm that CCA is not excessively sensitive to our particular choice of input data. The trial-shuffling test for significance and our inactivation experiments in Figures 7 and 8 are key reasons we believe that the moment-by-moment correlations identified by our CCA method are meaningful.

3) Have you considered that finding the optimal cross-area dimensions may require a time lag between areas? Presumably this could be answered by inspecting the pair-wise cross-correlograms you calculated.

We thank the reviewer for this suggestion. In terms of single-unit cross-correlations, we found that many neuron pairs had highest cross-correlation when the M2 neuron lead, but across the population as a whole, hierarchical bootstrapping found that the optimal timelag for pairwise cross-correlation was not significantly different from zero. As an additional way to investigate timing, we fit CCA models for data binned at 50, 75, and 100ms, and with lags from -500ms to +500ms. For 6 of 8 datasets, using 100ms time bins with no lag provided the highest correlation between M1 and M2 neural activity (see Methods lines 582-603). We think these analyses support our conclusion that the most correlated *population signals* between the regions do not

have a large timelag between M2 and M1. However, we wish to acknowledge that our population signals are computed off of very sparse samplings of neurons from each region. Given that single-synapse connections exist between M2 and M1, it is likely that a more complete sampling of M2 and M1 signals would reveal shared activity between M2 and M1 neurons with small time lags. Additionally, in Figure 6, we highlight that the local M1 and M2 population signals which are less correlated between the regions have a significant timelag in which M2 precedes M1.

Very minor comment:

- In some cases there are spaces between words and the corresponding reference number, no doubt a remnant of submission to a journal that used a different reference style.

We thank the reviewer for their keen eye! We found several instances of this stylistic error and fixed them.

References

Kornblith, S., Norouzi, M., Lee, H., and Hinton, G. (2019). Similarity of Neural Network Representations Revisited.

Makino, H., Ren, C., Liu, H., Kim, A.N., Kondapaneni, N., Liu, X., Kuzum, D., and Komiyama, T. (2017). Transformation of Cortex-wide Emergent Properties during Motor Learning. *Neuron* 94, 880-890.e8.

Perich, M.G., Gallego, J.A., and Miller, L.E. (2018). A Neural Population Mechanism for Rapid Learning. *Neuron* 100, 964-976.e7.

Reviewers' Comments:

Reviewer #1:

Remarks to the Author:

I thank the authors for their clear responses to my previous comments on their interesting manuscript. I only have a few minor moderate concerns/comments:

- Line: "computational modeling"? -> "computational methods"

- Lines 56-57: "This approach (...) dismisses " -> "May dismiss". Note that in a few of the studies cited by the authors (Perich et al) they could predict M1 latent activity from PMd (M2) latent activity, even during largely variable force field adaptation trials (both using PCs and single neuron activity). In my view, the result by Perich et al suggests that CCA between area-specific neural populations may not be that different from doing PCA within each area and building a model that predicts M1 from M2 (of course this could be tested on the authors' data but I don't think it's critical).

- Lines 83-84: "Together, our results indicate that cross-area M2-M1 population dynamics represent skilled motor learning." -> I'd say that M2-M1 interaction represents a component (or a "necessary" component) of skilled motor learning, but not the only one — we know that many other areas are involved in the execution of learnt skills. The fact that inactivating M2 prevents accurate movement execution could be due to eliminating M1's main inputs, which is a rather extreme perturbation from a physiological point of view.

- R2 of CCs. Thank you for your explanation. Just to make sure that I understood what the authors did: you split the data into 10 folds, and computed CCA between M2 and M1 using 9 of them. Then you took the n vectors found using CCA (where n is $\min(\text{neurons M1}, \text{neurons M2})$) and projected onto them the data for the corresponding area during the "test fold". What you plot is the square of the correlation that you get for that fold. Correct?

- Figure S3d. Thank you for including this. I think it'd be useful to see a clearer comparison between pre-learning CCs and to post-learning CCs, e.g., by dividing the latter trace by the former. This would give a good sense of how cross-area interactions become "stronger" after learning

- Figure 3: Is M1 Cross-area activity the projection of the M1 activity onto the CC vector, and M2 cross-area activity the same for M2?

Reviewer #2:

Remarks to the Author:

The ms has been adequately revised taking into account reviewers' suggestions. I have no issues regarding this revised version.

Reviewer #3:

Remarks to the Author:

In my opinion the authors thoroughly and thoughtfully addressed my comments, as well as those of the other reviewers. I have no further comments!

REVIEWERS' COMMENTS

Reviewer #1 (Remarks to the Author):

I thank the authors for their clear responses to my previous comments on their interesting manuscript. I only have a few minor moderate concerns/comments:

- Line: “computational modeling”? -> “computational methods”

We thank the reviewer for this comment. We have incorporated the suggested language in line 24.

- Lines 56-57: “This approach (...) dismisses “ -> “May dismiss”. Note that in a few of the studies cited by the authors (Perich et al) they could predict M1 latent activity from PMd (M2) latent activity, even during largely variable force field adaptation trials (both using PCs and single neuron activity). In my view, the result by Perich et al suggests that CCA between area-specific neural populations may not be that different from doing PCA within each area and building a model that predicts M1 from M2 (of course this could be tested on the authors’ data but I don’t think it’s critical).

We thank the reviewer for this comment. We have changed our text (“potentially dismisses”) to the suggested language (“may dismiss”) in line 57.

- Lines 83-84: “Together, our results indicate that cross-area M2-M1 population dynamics represent skilled motor learning.” -> I’d say that M2-M1 interaction represents a component (or a “necessary” component) of skilled motor learning, but not the only one — we know that many other areas are involved in the execution of learnt skills. The fact that inactivating M2 prevents accurate movement execution could be due to eliminating M1’s main inputs, which is a rather extreme perturbation from a physiological point of view.

We thank the reviewer for this comment. We have incorporated the suggested language in line 85.

- R2 of CCs. Thank you for your explanation. Just to make sure that I understood what the authors did: you split the data into 10 folds, and computed CCA between M2 and M1 using 9 of them. Then you took the n vectors found using CCA (where n is min(neurons M1, neurons M2)) and projected onto them the data for the corresponding area during the “test fold”. What you plot is the square of the correlation that you get for that fold. Correct?

We thank the reviewer for this question. That is correct! We believe this is equivalent to our calculation, which is ($R^2 = 1 - \text{sum squared error} / \text{total sum of squares}$). The example values plotted in Figure 2d are the average across all 10 combinations of testing/training data for each CV.

- **Figure S3d. Thank you for including this. I think it'd be useful to see a clearer comparison between pre-learning CCs and to post-learning CCs, e.g., by dividing the latter trace by the former. This would give a good sense of how cross-area interactions become "stronger" after learning**

We thank the reviewer for this suggestion. Please see below a figure of post-learning CCs divided by pre-learning CCs for M1 and M2 in the example animal from S3c. To calculate the modulation ratio, we first normalized each of the pre- and post-learning traces to the [0 1] range. However, we respectfully point out that these comparisons of mean activity collapse activity from trials (i.e. collapse a range of reach-to-grasp speeds); this is why we favored calculating movement-related activity on a single-trial basis in our manuscript.

- **Figure 3: Is M1 Cross-area activity the projection of the M1 activity onto the CC vector, and M2 cross-area activity the same for M2?**

We thank the reviewer for this question. That is correct!

Reviewer #2 (Remarks to the Author):

The ms has been adequately revised taking into account reviewers' suggestions. I have no issues regarding this revised version.

We thank the reviewer for this generous evaluation.

Reviewer #3 (Remarks to the Author):

In my opinion the authors thoroughly and thoughtfully addressed my comments, as well as those of the other reviewers. I have no further comments!

We thank the reviewer for this generous evaluation.